# Optimization of carbon and energy utilization through differential translational efficiency

Mahmoud M. Al-Bassam[1], Ji-Nu Kim[1], Livia S. Zaramela[1], Benjamin P. Kellman [1,2], Cristal Zuniga[1], Jacob M. Wozniak[3,4], David J. Gonzalez[3,4] & Karsten Zengler [1,5]

Control of translation is vital to all species. Here we employ a multi-omics approach to decipher condition-dependent translational regulation in the model acetogen *Clostridium ljungdahlii*. Integration of data from cells grown autotrophically or heterotrophically revealed that pathways critical to carbon and energy metabolism are under strong translational regulation. Major pathways involved in carbon and energy metabolism are not only differentially transcribed and translated, but their translational efficiencies are differentially elevated in response to resource availability under different growth conditions. We show that translational efficiency is not static and that it changes dynamically in response to mRNA expression levels. mRNAs harboring optimized 5′-untranslated region and coding region features, have higher translational efficiencies and are significantly enriched in genes encoding carbon and energy metabolism. In contrast, mRNAs enriched in housekeeping functions harbor suboptimal features and have lower translational efficiencies. We propose that regulation of translational efficiency is crucial for effectively controlling resource allocation in energy-deprived microorganisms.

[1] Department of Pediatrics, Division of Host−Microbe Systems and Therapeutics, University of California San Diego, 9500 Gilman Drive, La Jolla, CA 92093, USA. [2] Bioinformatics and Systems Biology Program, University of California, La Jolla, CA 92093, USA. [3] Department of Pharmacology, University of California San Diego, La Jolla, CA 92093, USA. [4] Skaggs School of Pharmacy and Pharmaceutical Sciences, University of California San Diego, La Jolla, CA 92093, USA. [5] Center for Microbiome Innovation, University of California San Diego, La Jolla, CA 92093, USA. Correspondence and requests for materials should be addressed to K.Z. (email: kzengler@ucsd.edu)

The metabolic versatility of acetogens for the fermentation of a large number of sugars yields great promise for the production of biofuels and commodity chemicals. In particular the ability to grow autotrophically with $H_2$:$CO_2$ or syngas ($H_2$/CO/$CO_2$) makes these organisms an ideal chassis for sustainable bioproduction and acetogenic clostridia are currently deployed for the commercial conversion of syngas to biofuels. *Clostridium ljungdahlii* is emerging as a promising cell factory for bioproduction[1] as well as a model organism for gaining in-depth knowledge necessary to develop new design strategies for acetogens. *C. ljungdahlii* is readily cultured heterotrophically in the laboratory in simple media, either on a diverse set of five or six carbon sugars, or autotrophically with CO or $H_2$ as electron donor. Furthermore, metabolic models and genetic manipulation tools already developed and optimized for this organism make *C. ljungdahlii* an ideal candidate for the study of acetogenesis[2–4]. However, in order to harness the full biosynthetic potential, it is important to understand the regulatory mechanisms that orchestrate energy metabolism in *C. ljungdahlii*. These include, but are not limited to, the Wood−Ljungdahl pathway (WLP), the formate dehydrogenase complex, the hydrogenase complex, and the Rnf complex, all central to energy equilibrium in *C. ljungdahlii*[5–7]. A thorough understanding of all factors that regulate energy metabolism under autotrophic and heterotrophic growth conditions is crucial for the metabolic engineering of acetogens and for optimizing targeted production of desired chemicals.

In recent years, next-generation omic approaches, such as RNA-seq, proteomics, and metabolomics, have been employed to identify the functionality and organizational structure of acetogenic bacterial genomes[8–11]. These approaches directly addressed the genotype−phenotype relationship in bacteria, providing crucial insights into the design strategies for microbial cell factories. Ribosome profiling (Ribo-seq) has recently enabled the determination of the numbers and locations of ribosomes on mRNAs in vivo[12] and in combination with RNA-seq, has facilitated the global measurement of translational efficiency (TE) and thus provided new insights into translational regulation[13,14]. Translation is a major energy burden especially for cells growing in nutrient-deficient conditions. In these niches optimization of resource allocation becomes increasingly critical for survival. We combined Ribo-seq, RNA-seq, and transcription start site sequencing (TSS-seq) to study how resources are allocated under energy-rich heterotrophic growth and energy-deprived autotrophic growth conditions in the model acetogen *C. ljungdahlii*. We provide evidence that metabolic pathways involved in carbon and energy metabolism are strongly regulated at the translational level. We find that RAST subsystems belonging to these pathways are significantly enriched in mRNA with optimized 5′-untranslated region (5′UTR) features (i.e. high-affinity ribosome-binding site (RBS), increased AU content upstream of the RBS, and optimal distance of RBS from the translation initiation site) and optimized coding region features (high codon adaptation index (CAI) and low AU content). Optimization of mRNA features increases the affinity of mRNA to ribosomes and hence facilitates the proportional increase of TE in response to increased mRNA expression. In contrast, we find that mRNAs belonging to genes involved in housekeeping functions are significantly enriched with suboptimized features that reduce mRNA affinity to ribosomes. We propose that selective control of TE in key metabolic and energy pathways is critical for thriving in nutritionally deprived niches.

## Results

### Multi-omics analyses of hetero- and autotrophic cells.
We carried out RNA-seq and Ribo-seq experiments for autotrophic cultures of *C. ljungdahlii* grown either on CO or $H_2$:$CO_2$ and heterotrophic cultures grown on fructose. To enable direct comparison between transcription and translation, strand-specific RNA-seq libraries were prepared from the same lysates used for Ribo-seq experiments in biological duplicates. RNA-seq and Ribo-seq libraries were deeply sequenced (Supplementary Fig. 1) and mapped reads were normalized as FPKM (fragments per kilobase per million) and RPKM (reads per kilobase per million), respectively. RNA-seq and Ribo-seq replicates for cultures grown on CO, $H_2$:$CO_2$, or fructose were highly reproducible with Pearson's correlations for RNA-seq = 0.995, 0.991, and 0.989, respectively and Pearson correlations for Ribo-seq = 0.995, 0.952, and 0.926, respectively (Supplementary Fig. 2a). Whereas Spearman correlations for Ribo-seq were 0.99, 0.95, and 0.97 for CO, fructose, and $H_2$:$CO_2$, respectively. At the RAST subsystem level (see below) the Pearson's correlations for Ribo-seq are 1.00, 0.97, and 0.97 for CO, fructose, and $H_2$:$CO_2$, respectively (Supplementary Fig. 2b). Whereas Spearman correlations were 1.00, 0.99, and 0.99 for CO, fructose, and $H_2$:$CO_2$, respectively. The high correlations between biological replicates reflects the high reproducibility of our data.

While the majority of genes are regulated at the transcriptional level, transcription and translation in bacteria are spatially coupled[15] and many genes are subjected to firm translational control[13,16,17]. In line with previous findings in *Escherichia coli*[14] and *Streptomyces coelicolor*[18], RNA-seq and Ribo-seq data from *C. ljungdahlii* were moderately correlated in all conditions tested (Fig. 1a), hinting at widespread translational regulation. We thus calculated the TE of each gene by dividing the translational level (Ribo-seq RPKM) by the transcriptional level (RNA-seq FPKM) and noticed significant discrepancy in TE among different genes (Fig. 1b).

### Differential translation reveals overall resource allocation.
Classification of genes into discrete functional units and the measurement of transcription or translation of these units provide valuable insight into how resources are allocated to each function. Therefore, we functionally annotated the *C. ljungdahlii* genome using RAST[19], resulting in the classification of 1731 genes into 270 subsystems (Supplementary Data 1). Differentially translated subsystems under all growth conditions were determined by DESeq2[20]. To assess differential translation and transcription across the three conditions, RNA-seq and Ribo-seq data for growth on CO, $H_2$:$CO_2$, and fructose were normalized as percent values and compared at the RAST subsystem level (Supplementary Fig. 3; see Methods). The top 20 differentially translated subsystems (DESeq2 $P < 0.01$) under heterotrophic (Fig. 2, top) and autotrophic conditions (Fig. 2, bottom) are shown. The top 20 differentially translated subsystems in heterotrophic and autotrophic conditions were highly associated with carbon and energy sources present in the corresponding growth media.

In heterotrophic growth, three differentially upregulated subsystems were related to carbon metabolism (Fig. 2: H1, H2, and H7) and 13 were related to de novo macromolecule synthesis and maintenance (Fig. 2: H3, H5, H6, H9−H16, H18, and H19). The remaining four clusters (Fig. 2: H4, H8, H17, and H20) had no obvious link to heterotrophic metabolism or fast growth. Glycolysis (H1) and the pentose phosphate pathways (H2) were the top differentially upregulated subsystems followed by the chorismate synthesis subsystem (H3), which is the precursor molecule for de novo synthesis of the aromatic amino acids phenylalanine, tyrosine, and tryptophan. The sporulation cluster (H4) was unexpectedly highly upregulated. After close inspection, we found that out of four genes in this subsystem, *Clju_c41620* (encoding a putative RNA-binding S1 domain-containing protein) was the only differentially translated gene

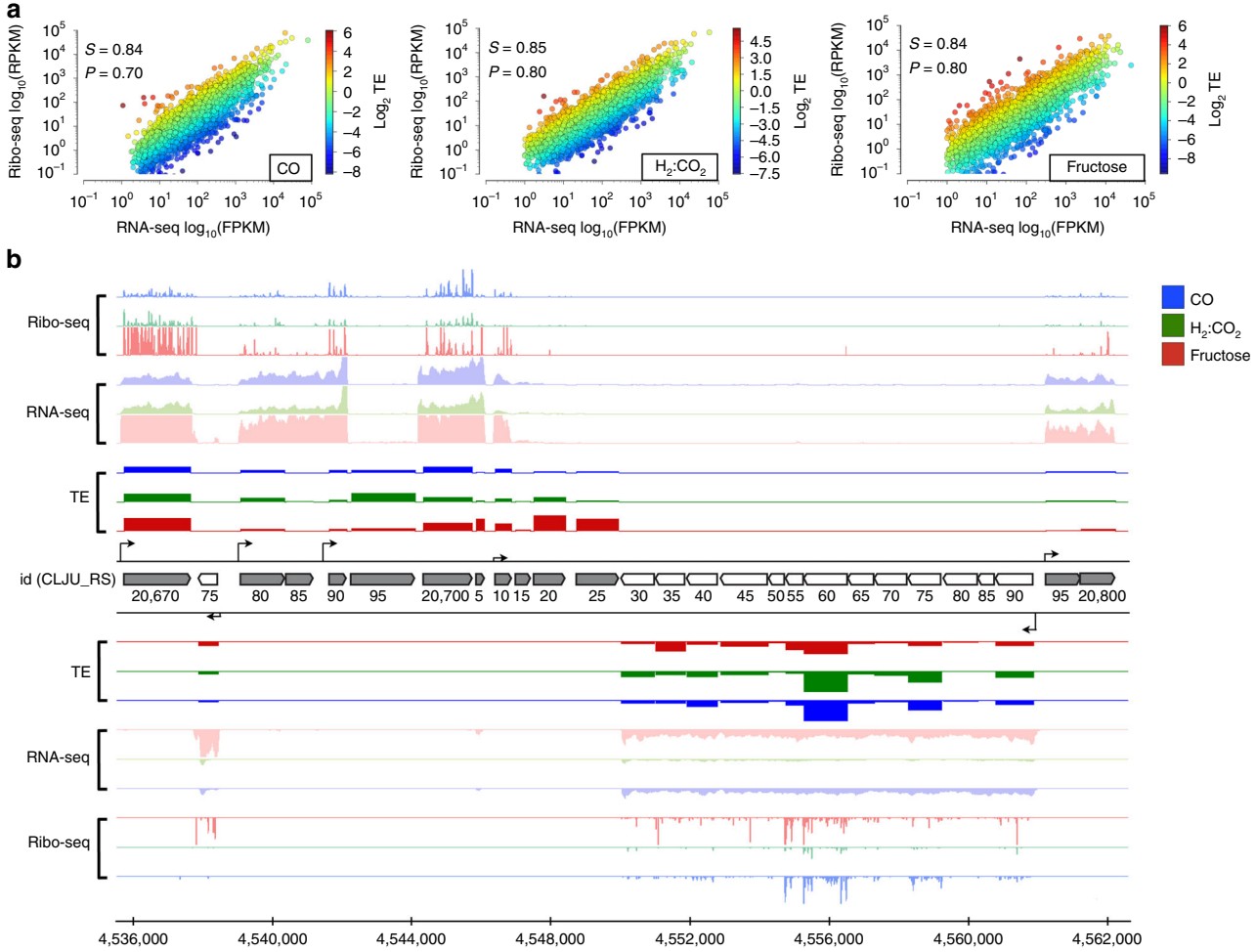

**Fig. 1** Overview of omic experiments carried out and the correlation between RNA-seq and Ribo-seq in all growth conditions. **a** Correlations between RNA-seq and Ribo-seq in CO, $H_2:CO_2$, and fructose. Pearson's (*P*) and Spearman's (*S*) correlation coefficients are shown inside each subfigure. Colors in the scatter plot represent the translational efficiency values. The color bars show TE $\log_2$ values. **b** An example of Ribo-seq, RNA-seq, TE, and TSS profiles mapped onto genomic region between 4,535,800 to 4,564,000 and showing genes *Clju_RS20670* to *Clju_RS20800*. RNA-seq and Ribo-seq profiles were normalized to reads per million (RPM). TE of each gene is a ratio Ribo-seq to RNA-seq level. Arrows indicate transcription start site (TSS) positions

(Supplementary Data 2). This protein weakly interacts with the ribosome and facilitates the recognition of the translation initiation site (further discussed below). The differential translational upregulation of the initiation factors (IF1, IF2, and IF3, Fig. 2: H12) is consistent with higher growth rate under heterotrophic growth. It was previously reported that the transcription of IF3 does not vary with different growth rates in *E. coli*; however, quantitative analysis demonstrated that IF3 abundance vary in parallel to the ribosomal levels[21]. However, these three IFs had considerably inefficient translation (see discussion). Overall, the differentially translated subsystems under heterotrophic growth are consistent with functions required for fast growth and effective utilization of fructose.

In autotrophic growth, eight subsystems were closely related to carbon fixation and energy conservation (Fig. 2: A1, A4, A6−A8, A12, A14, and A18), two subsystems were related to fermentation (Fig. 2: A2 and A5) and three subsystems were related to motility (Fig. 2: A3, A9, and A17). The top four translationally upregulated subsystems (A1−A4) consisted of the carbon monoxide dehydrogenase (CODH)/AscA cluster, 2,3-butanediol dehydrogenase (BDD), flagellum, and the Rnf complex. The CODH/AscA complex is directly involved in carbon fixation and

energy conservation through the Wood−Ljungdahl pathway (WLP). Remarkably, BDD translation represented 7% of the total translation under CO growth. The flagellum, flagellar motility, and bacterial chemotaxis represented 4% of total translation. Cells growing on CO were conspicuously the most motile when examined under the microscope, which supports the observed differential translation.

Differentially translated subsystems specific to each of the two autotrophic conditions (CO and $H_2:CO_2$) mostly involved energy utilization and carbon fixation (Supplementary Fig. 4). The Rnf complex, flavodoxin, and the aldehyde:ferredoxin oxidoreductase were significantly upregulated under $H_2:CO_2$ growth compared to growth under CO. Whereas the translation of acetoin/2,3-butanediol metabolism, the flagellum, and one-carbon metabolism (i.e. WLP) was differentially upregulated in CO compared to $H_2:CO_2$. Thus, reflecting intricate metabolic adjustments required for optimum utilization of the energy and carbon sources in CO and $H_2:CO_2$.

Interestingly, the majority of translationally upregulated subsystems show differential increase in TE. In heterotrophic growth, most subsystems showed differential increase in TE compared to autotrophic growth (up-pointing triangles in Fig. 2),

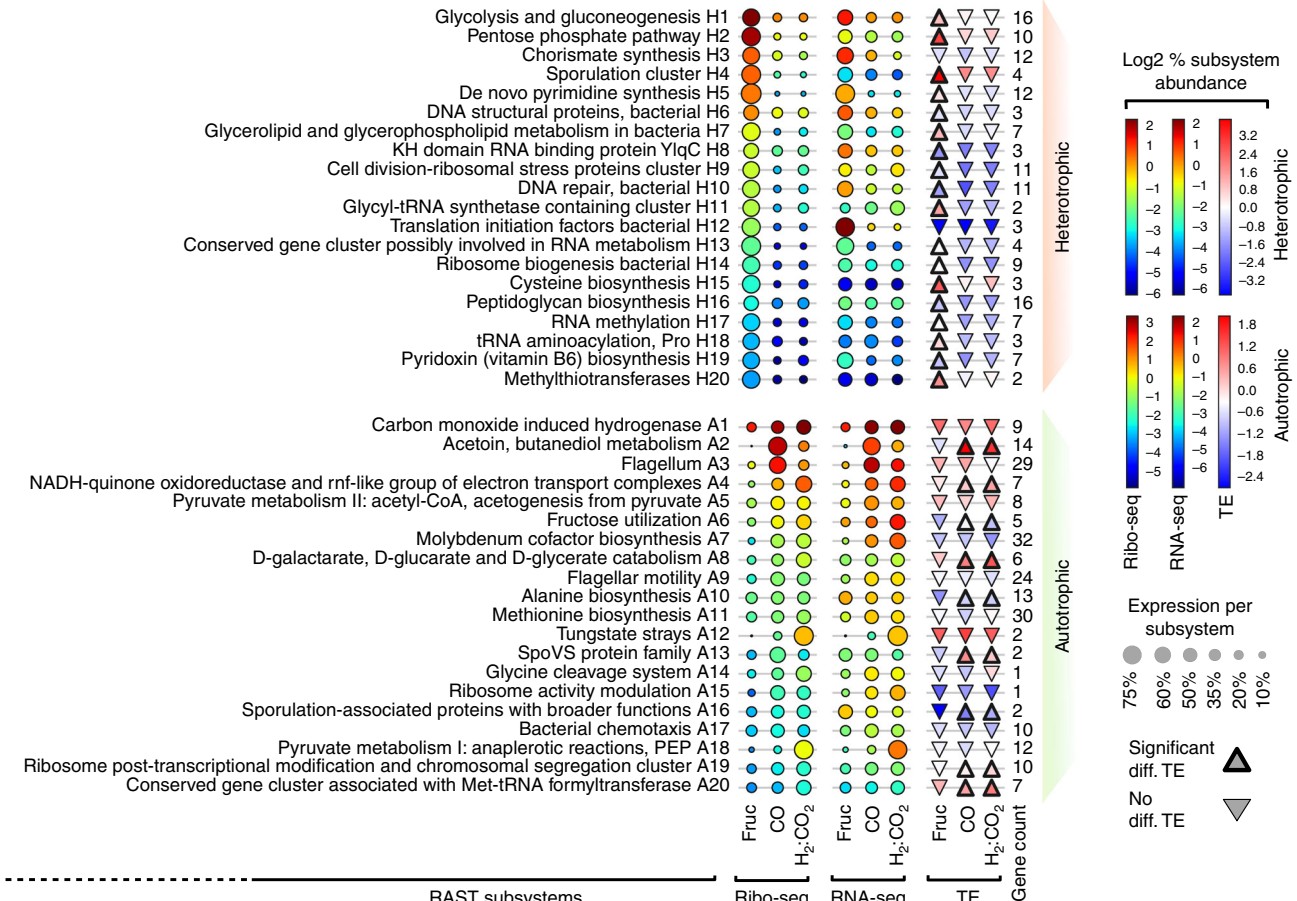

**Fig. 2** Differential translation and differential TE of subsystems in fructose, CO and $H_2$:$CO_2$ growth cultures. Genes were grouped into RAST subsystems and translation and transcription were both percent-normalized per each experiment. Each column represents the level of translation or transcription (% normalized) per each experiment and the bubble color reflects the intensity (% normalized). Each row represents a comparison between the three datasets and the size of bubbles represents the level of translation or transcription per each subsystem (% normalized). TE is compared per each subsystem and depicted by triangles. Subsystems with differentially increased TE are depicted by up-pointing triangle with thick edges and the colors represent percent-normalized values across the three conditions. The number of genes is shown per each subsystem. The top panel manifests the top 20 subsystems that are differentially upregulated (*P* < 0.01) at the translational level in fructose (heterotrophic) relative to both autotrophic conditions and the data are sorted according to percent translation in fructose. The bottom panel manifests subsystems that are differentially upregulated (*P* < 0.01) at the translational level in both autotrophic conditions with data sorted in descending order according to percent translation levels in CO. All values are shown as $\log_2$

most notably the glycolysis/gluconeogenesis and the pentose phosphate pathway (Fig. 2, H1 and H2, respectively). Subsystems with differential TE were less frequent under autotrophic growth. However, the 2,3-butanediol/acetoin fermentation pathway and the Rnf complex cluster (Fig. 2, A2 and A4, respectively) showed the most differential TE. These findings demonstrate a tight link between TE and the growth condition-specific metabolic and energy demands.

**Differential translation at the gene level**. To gain insight into how TE is differentially controlled under autotrophic and heterotrophic conditions, we analyzed the genes comprising the major carbon and energy subsystems that were significantly enriched (Fig. 2). Specifically, we discuss genes comprising the glycolysis/gluconeogenesis pathway, the WLP, all fermentation pathways, the Rnf complex, and the ATPase complex (Fig. 3). Genes with redundant functions, which are not differentially translated, were not included in the analysis. As expected, the majority of genes in glycolysis/gluconeogenesis were differentially enriched during heterotrophic growth (Fig. 3, blue arrows), whereby fructose is taken up preferentially via the fructokinase/fructose-6-phosphate isomerase (G1) and the 6-

phosphofructokinase (G3) route. Under autotrophic growth, the fructose phosphotransferase system (PTS) and 1-phosphofructokinase (G2) were also significantly enriched. Two enzymes involved in pyruvate metabolism were differentially translated (P4 and B1 in Fig. 3). The incomplete TCA cycle exhibited differential translation, whereas genes involved in fermentation were only differentially translated under autotrophic growth. Most notable are E1 (bifunctional aldehyde/alcohol dehydrogenase) and B3 (2,3-butanediol dehydrogenase), both differentially translated with high efficiency under autotrophic conditions (Fig. 3, A2 in Fig. 2). The WLP is mostly differentially translated under autotrophic growth with W5 (methenyl-THF cyclohydrolase) and W7 (methylene-THF reductase) being the least efficient (Fig. 3). Interestingly, W6 (Methylene-THF reductase) is not differentially translated under any of the growth conditions. It has been shown that NADPH and adeno-sylmethionine allosterically regulate the activity of methylene-THF reductase (W6) in a mutually antagonistic manner[22,23], which could imply that the activity of this enzyme is not merely governed by its abundance. All genes encoding the F1F0 ATPase are differentially transcribed (Supplementary Data 2) and differentially translated under heterotrophic growth condition. The

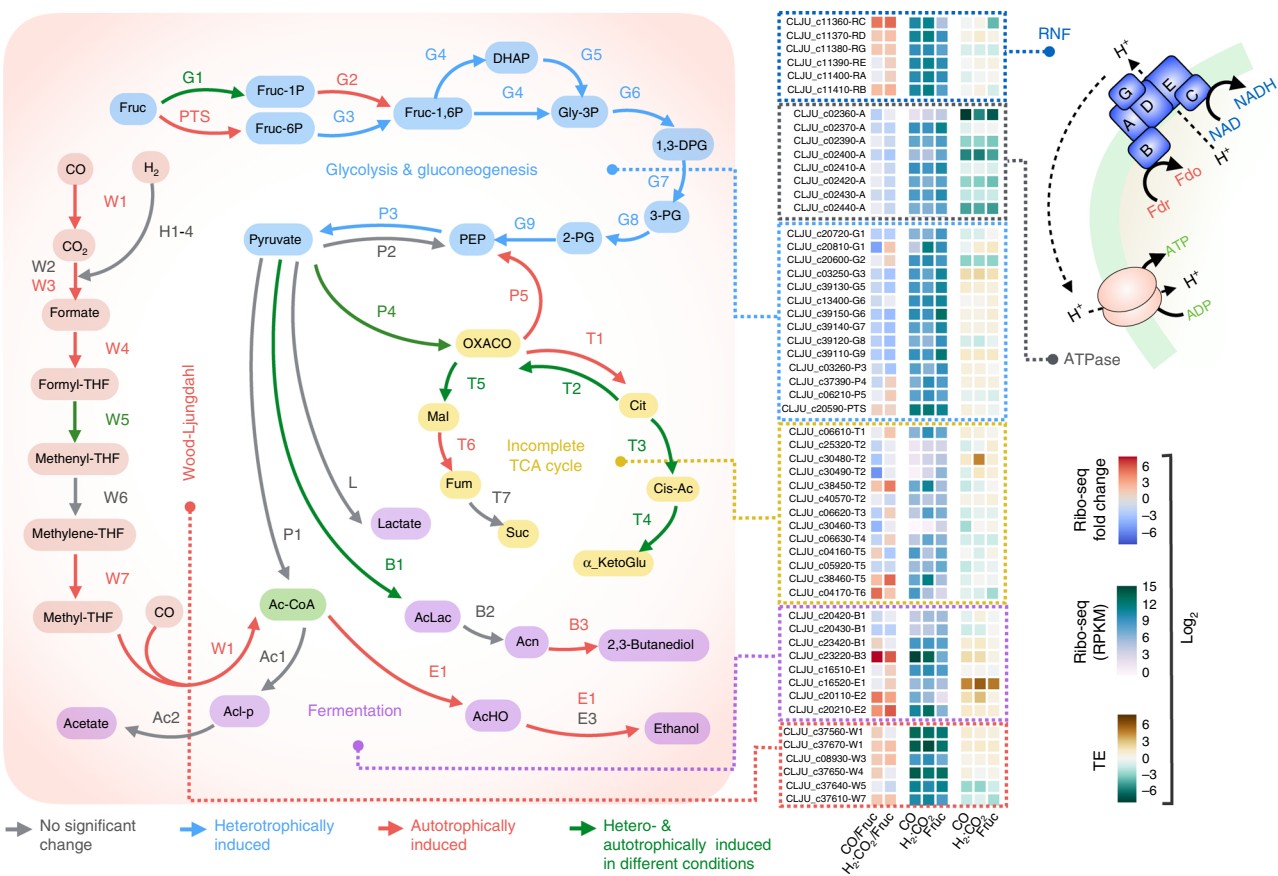

**Fig. 3** Metabolic map of major carbon and energy pathways exhibiting differential translation and differential TE. Differential fold change is calculated as the log$_2$ CO/fructose or H$_2$:CO$_2$/fructose translation ratios. Heterotrophically induced (red arrows), autotrophically induced (blue arrows), insignificant (gray arrows) and condition-specific (green arrows) translation is depicted in all pathways. Glycolysis and Gluconeogenesis: fructose phosphotransferase system (PTS); fructokinase/fructose-6-phosphate isomerase (G1); 1-phosphofructokinase (G2); 6-phosphofructokinase (G3); ketose-bisphosphate aldolase (G4); triose-phosphate isomerase (G5); glyceraldehyde-3-phosphate dehydrogenase (G6); phosphoglycerate kinase (G7); phosphoglycerate mutase (G8); enolase phosphopyruvate hydratase (G9); pyruvate:ferredoxin oxidoreductase (P1); pyruvate, phosphate dikinase (P2); pyruvate kinase (P3); pyruvate carboxylase (P4); PEP carboxykinase (P5). Fermentation: phosphotransacetylase (Ac1), acetate kinase (Ac2), bifunctional aldehyde/alcohol dehydrogenase (E1), aldehyde:ferredoxin oxidoreductase (E2), additional alcohol dehydrogenases (E3), acetolactate synthase (B1), acetolactate decarboxylase (B2), 2,3-butanediol dehydrogenase (B3); lactate dehydrogenase (L). Incomplete TCA cycle: citrate synthase (T1); citrate lyase (T2); aconitase (T3); isocitrate dehydrogenase (T4); malate dehydrogenase (T5); fumarase (T6); fumarate reductase (T7). Wood−Ljungdahl pathway: electron-bifurcating [FeFe] hydrogenase (H1); other [FeFe] hydrogenases (H2); [NiFe] hydrogenase (H3); hydrogenase maturation factor (H4); bifunctional CO dehydrogenase/ acetyl-CoA synthase (CODH/ACS) (W1); seleno formate dehydrogenase (W2); non-seleno formate dehydrogenase (W3); formyl-THF ligase (W4); methenyl-THF cyclohydrolase (W5); methylene-THF dehydrogenase (W6); methylene-THF reductase (W7). Rnf complex and ATPase: RnfC (RC); RnfD (RD); RnfG (RG); RnfE (RE); RnfA (RA); RnfB (RB); ATPase (A). Fructose (Fruc); fructose 1-phosphate/6-phosphate (Fruc-1P/-6P); fructose 1,6-bisphosphate (Fru-1,6P); dihydroxyacetone phosphate (DHAP); glycerol 3-phosphate (Gly-3P); 1,3-bisphosphoglycerate (1,3-DPG); 3-phosphoglycerate (3-PG); 2-phosphoglycerate (2-PG); phosphoenolpyruvate (PEP); oxaloacetate (OXACO); citrate (Cit); isocitrate (Cit-Ac); a-ketoglutarate (a-KetoGlu); malate (Mal); fumarate (Fum); succinate (Suc); acetolactate (AcLac); acetoin (Acn); acetaldehyde (AcHO); acetyl-phosphate (Acl-p); tetrahydrofolate (THF); reduced ferredoxin (Fdr); oxidized ferredoxin (Fdo)

remarkable low-TE of the ATPase cluster implies that its translation is relatively more resilient to transcriptional fluctuations. The Rnf genes (*rnfCDGEAB*) are differentially transcribed, differentially translated, and most genes, including the Rnf regulator *rseC*, exhibit differential TE under autotrophic growth conditions (further discussed below).

**The Rnf complex is under strict translational control.** The *rnfC* gene is transcribed at a significantly lower level during heterotrophic growth (FPKM = 621.7 under fructose growth compared to 2560.7 and 3080.8 for CO and H$_2$:CO$_2$ growth, respectively; Figs. 3, 4a). Notably, *rnfC* is acutely translationally repressed under heterotrophic condition (TE = 0.1 for fructose compared

to 0.9 for CO and 1.3 for H$_2$:CO$_2$; Fig. 4b, Supplementary Data 2), thus contributing only ~1% of the entire Rnf complex translation. Under heterotrophic growth, the Rnf regulator *rseC* is transcribed at a high level in each growth condition (FPKM = 2098.1, 2991.8, 1022.9, for CO, H$_2$:CO$_2$, and fructose growth conditions, respectively). However, *rseC* translation is also highly repressed under heterotrophic growth at the translational level comparable to that of *rnfC* (TE = 2.0, 2.2, 0.3 for CO, H$_2$:CO$_2$, and fructose growth conditions, respectively; Fig. 4b). Ribo-seq results were consistent with relative protein levels of RscE, RnfC, RnfD, RnfG, and RnfB (Fig. 4c, Supplementary Table 1). RnfE showed slight increase under fructose relative to H$_2$:CO$_2$. RnfA was not detected in our proteomics experiment. These results corroborate our finding that the Rnf complex (in particular *rnfC* and

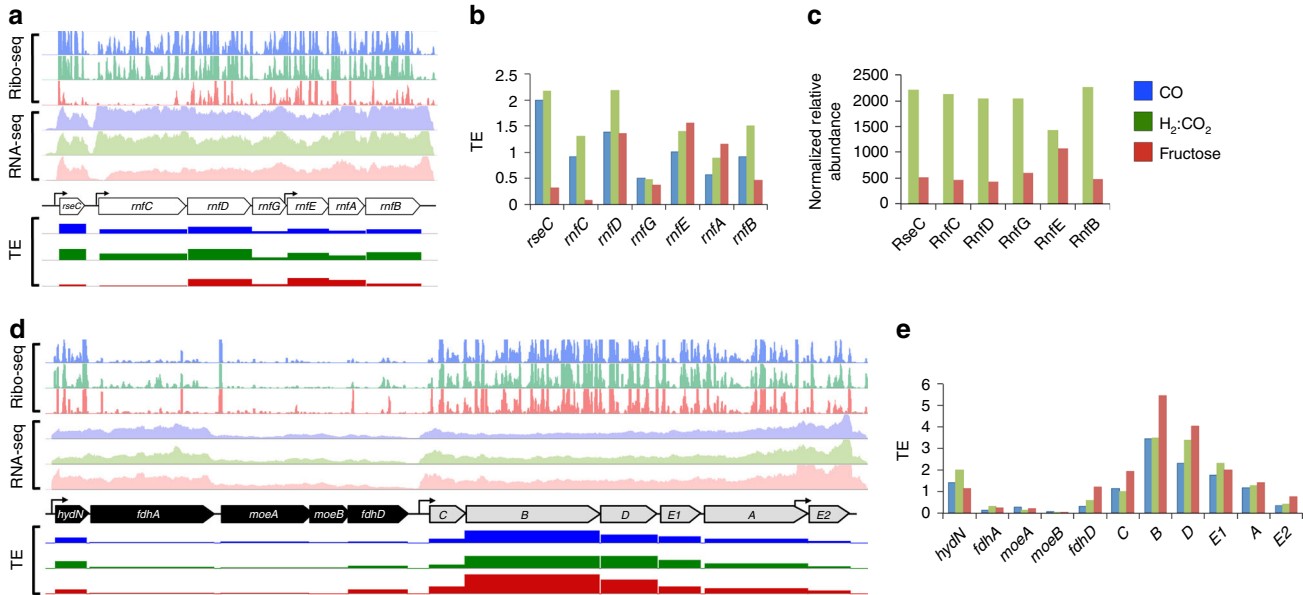

**Fig. 4** Transcriptional and translational regulation of the Rnf (white), formate dehydrogenase (black), and hydrogenase (gray) complexes in all growth conditions. Results are manifested for CO, $H_2:CO_2$, and fructose in blue, green and red, respectively. **a** Transcription and translation of the Rnf complex (normalized to reads per million (RPM)). The *rnf* complex (*Clju_c11350-Clju_c11410*) has one major TSS upstream of *rnfC*. The *rnfEAB* genes are transcribed from an internal promoter that is positioned at the 3′-end of *rnfG*. The *rseC* gene is transcribed from one TSS and transcription is comparable across all conditions; however, it is poorly translated in fructose. **b** Comparison between TE of the Rnf genes in each condition. The *rnfC* gene has the lowest TE in fructose. **c** Shotgun proteomics showing the relative abundance of the Rnf complex proteins (RnfA was not detected) in $H_2:CO_2$ and fructose. **d** Expression of the formate dehydrogenase and the hydrogenase genes (*Clju_c06990-Clju_c07080*, normalized to reads per million (RPM)). Both clusters are expressed from upstream TSSs. The *hydN* and the *fdhA* genes are translated at a much lower efficiency compared to the hydrogenase B and D genes despite having higher transcription. **e** Comparison between the TE of the hydrogenase and the formate dehydrogenase genes in each condition. The hydrogenase *E2* gene is transcribed at a higher level from an internal promoter

*rseC*) is highly translationally repressed under heterotrophic growth.

**Drastic TE differences in two energy-metabolism operons**. *C. ljungdahlii* and *Clostridium autoethanogenum* are two phylogenetically indistinguishable species as they have >98% overall genome similarity[24,25]. The only active hydrogenase (Hyd) in *C. ljungdahlii* is the one orthologous to HytABCDE1E2 in *C. autoethanogenum*, which has been demonstrated to be the only active hydrogenase during $H_2:CO_2$ growth[26]. In *C. ljungdahlii*, Hyd catalyzes the reduction of NADP and ferredoxin and the oxidation of $H_2$ under $H_2:CO_2$ growth. Additionally, Hyd interacts with formate dehydrogenase (Fdh) and the resulting complex (Hyd-Fdh) catalyzes the reduction of $CO_2$ to formate and the oxidation of $H_2$[26] (Supplementary Fig. 5a). Under CO growth, the bifurcating carbon monoxide dehydrogenase (CODH) catalyzes the oxidation of CO to $CO_2$ and the reduction of ferredoxin. The Hyd-Fdh complex then catalyzes the oxidation of ferredoxin and the reduction of $CO_2$ to formate (Supplementary Fig. 5b). Under heterotrophic growth, the pyruvate ferredoxin oxidoreductase catalyzes the oxidation of pyruvate to acetyl-CoA, the reduction of ferredoxin and the generation of $CO_2$ as byproduct[7]. CODH catalyzes the oxidation of ferredoxin and the reduction of $CO_2$ into CO, whereas Hyd-Fdh catalyzes the reduction of $CO_2$ into formate using reduced ferredoxin (Supplementary Fig. 5c). Fdh and Hyd are both multimeric complexes, both active under all growth conditions tested, and both are essential for the WLP, which plausibly underscore the observed stable TE of both complexes under all conditions (Fig. 4d, e). Our omics analyses illustrate that at least *hydN* and *fdhA* are transcribed from one upstream TSS and their transcriptional levels are greater than

*hydCBDAE1*. The latter genes are also transcribed from one detectable TSS, whereas *hydE2* is transcribed from an internal TSS positioned at the 3′ end of *hydA* (Fig. 4d). Despite higher transcriptional levels of *hydN* and *fdhA*, the *hydBDE1* genes are translated at a much higher level (higher TE). In fact, *hydB* is at least threefold more translationally efficient than *hydN* and *fdhA* (Fig. 4e). These results suggest translational regulation is seminal for the regulation of key energy conservation centers in this model acetogen.

**Features in the 5′UTR region and the coding region govern TE.** The 5′UTR region is an important regulatory center for determining the fate of mRNA[27]. There are many regulators that directly act on the 5′UTR, resulting either in the reduction or the increase of TE. These include RNaseE[28], the global regulator CsrA[29] (see Discussion), the ribosomal protein S1 (RpsA)[30], and the translation initiation factors (IF1-3). RpsA (*Clju_c41620*), a ribosomal protein weakly associated with the 30S ribosomal subunit, has strong affinity towards AU-rich regions at the 5′UTR[31,32] and interacts with the 5′UTR of mRNA through a 10–15 nt motif to facilitate the initiation of translation[33]. In addition, RPS1 furnishes the 30S subunit with an RNA chaperone activity that is essential for the binding and unfolding of structured mRNAs, allowing the correct positioning of the initiation codon for translation[34]. Further, RPS1 competes with RNases for the binding of AU-rich regions, plausibly protecting AU-rich upstream RBS region (URR) from degradation[30], which leads to increased TE[31,35]. Overall, the AU content and the RBS sequence determine how these factors interact with the 5′UTR to control the efficiency of ribosome binding. Moreover, features present on the mRNA coding region can also have strong influence on TE, including

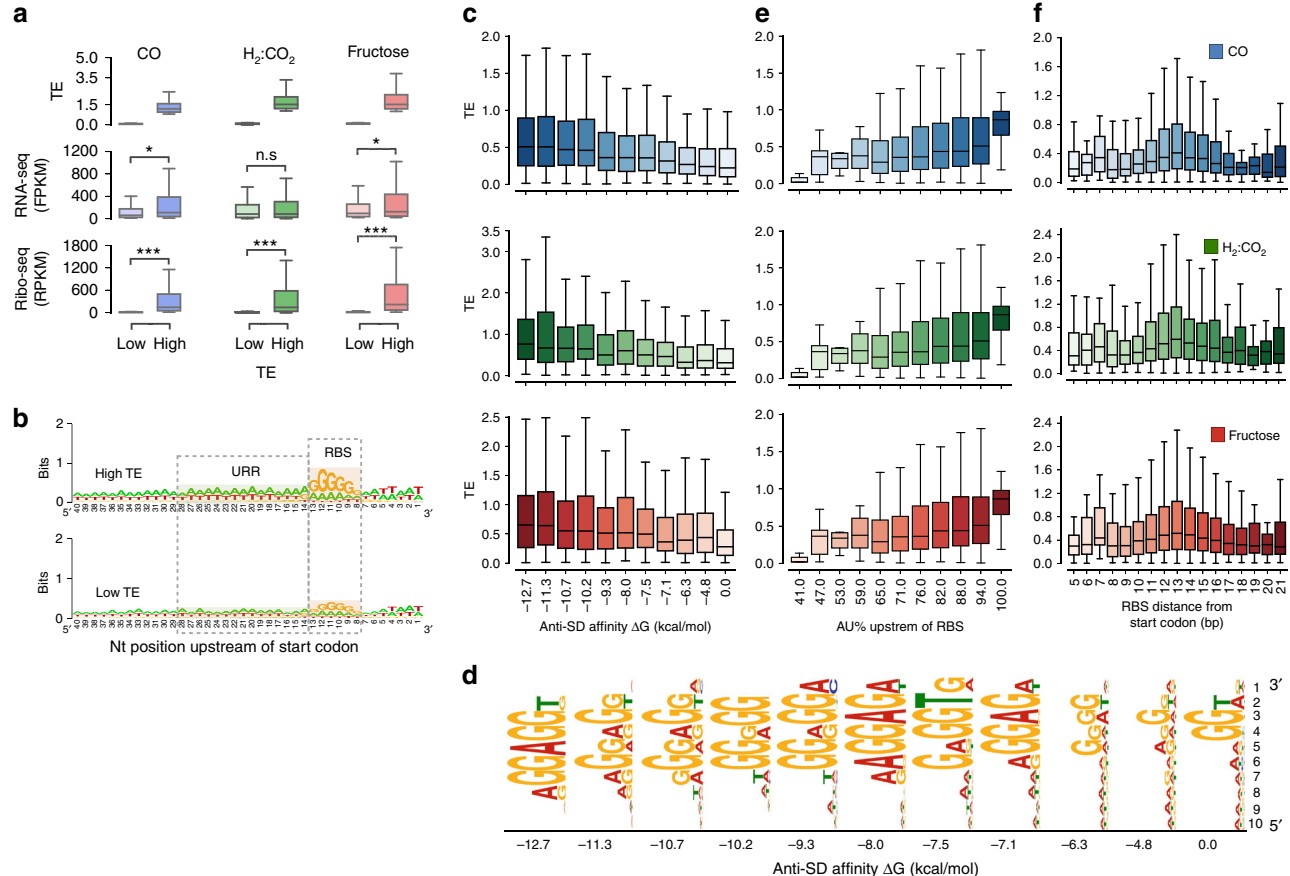

**Fig. 5** Influence of 5′UTR features on TE. **a** Comparison between genes with low and high TE in all conditions. Low-TE genes are below 20th percentile, whereas high-TE genes are above 80th percentile in all conditions. $P < 1e^{-10}$ are signified with "****", $1e^{-10} \leq P < 0.01$ are signified with "*" and $P \geq 0.01$ are signified with "n.s". **b** Low- and high-TE genes in all conditions have visible differences in their RBS sequence and the AU content in their upper RBS region (URR). The RBS and the URR regions are both highlighted with dotted gray boxes. **c** RBS affinity towards the anti-Shine Dalgarno (aSD) sequence (AAGGAGGU) positively affects TE in all conditions. The affinities of RBS towards the aSD region were categorized into 11 groups ranging from ΔG of 0 to −12.7. **d** The RBS motif per each category in CO was determined using MEME. **e** Positive effect of the 15 bp AU% content in the URR on TE in all conditions. TSS data were used to ensure that TSS is upstream of the URR. **f** The distance of the RBS 5′ end from the start codon is most optimum at 13 bp. Deviation of the RBS position in either direction negatively influences TE

tertiary mRNA structure and codon usage[36]. Therefore, it is vital to understand how these mRNA features influence TE.

To explore these features and determine their influence on translation and TE, we compared RNA-seq and Ribo-seq data of genes with low-TE (<20th percentile) or high-TE (>80th percentile). The difference between the two sets was strikingly more significant at the translational level (Wilcoxon signed-rank test $P = 1.7e^{-92}$, $1.2e^{-78}$, $8.2e^{-81}$ for CO, H₂:CO₂, and fructose, respectively) when compared to the transcriptional level (Wilcoxon signed-rank test $P = 1.2e^{-9}$, 0.13, $4.4e^{-6}$ for CO, H₂:CO₂, and fructose, respectively), implying strong translational regulation (Fig. 5a).

Previous studies have reported direct regulation of TE via the 5′UTR[37,38]. Here we investigated the effect of different features in the 5′UTR on TE. To accurately determine the 5′UTR regions, we first performed a comprehensive TSS analysis (Supplementary Data 3, Supplementary Fig. 6), using RNA extracted from four different growth conditions (see Methods). We determined a total of 1465 TSSs that correspond to the 5′-end of the primary transcriptome. The TSSs were further categorized by their genomic locations (Supplementary Fig. 6a). 1245 TSSs were annotated as primary TSSs (see Methods), which cover 29% of total gene content excluding operons and 50% of total gene

content including operons. In addition, we detected 116 internal TSSs and 25 antisense TSSs that could manifest potential control of gene expression[39]. One hundred and twenty-five orphan TSSs were also identified at intergenic regions with no associated genes, suggesting the presence of novel transcriptional units. Alignment of 50 bp upstream of TSSs revealed conservation of two motifs at −10 and to a lesser extent at −35 consistent with sigma factor binding motifs, implying high-accuracy detection of TSS (Supplementary Fig. 6b). Notably, we could not detect any leaderless genes under the growth conditions tested, which further emphasize the importance of translational regulation via the 5′UTR in *C. ljungdahlii*.

To investigate *cis*-acting regulatory elements of translational control, we defined the 5′UTR from the regions between primary TSSs and start codon of corresponding genes. The most frequent size range of 5′UTR distribution was 20–39 nt (Supplementary Fig. 6c). The median 5′UTR length was 47 nt, implying that for the vast majority of genes, *cis*-acting elements and secondary structures play a critical role in translational regulation. The RBS is one of the critical elements for translational initiation[40], which in turn directly impacts TE. We compared the composition of the −10 and −35 regions of the 5′UTR by analyzing 40 nt upstream of the start codon using WebLogo[41]. There were two clear

differences between low-TE and high-TE genes, namely the high-TE genes had a stronger RBS motif and the upper RBS region (URR) had an increased AU content (Fig. 5b). Based on these differences, we investigated how TE is influenced by (i) RBS affinity towards the anti-Shine Dalgarno (aSD) sequence (AAGGAGGU), (ii) the RBS distance from the TSS, and (iii) the AU% content of the URR. We measured the affinity of the aSD sequence towards RBS (see Methods) for both low- and high-TE genes. The difference was highly significant (Supplementary Table 2) between the two groups under all three conditions, suggesting that RBS affinity towards the initiating ribosomes is a key determinant for TE. Further, we organized all genes into 11 groups according to their ΔG of affinity and compared their TE (Fig. 5c). The gradual decrease in median TE with increasing ΔG implies that TE is strongly influenced by the RBS affinity towards the aSD (Fig. 5c). Furthermore, MEME analysis (http://meme-suite.org/tools/meme) showed that the RBS motif conservation increased with TE and those with lowest TE had a hardly recognizable RBS motif, whereas groups with high-TE exhibited optimal RBS motifs (Fig. 5d).

We reasoned that AU-rich URRs could result in greater TE. To test this hypothesis, we calculated the AU% in regions 15 nt (15 nt showed strongest difference between low- and high-TE sets in Fig. 5b) upstream of each RBS. To eliminate false positives arising from the high AT content of the *C. ljungdahlii* genome (31.1% GC), we limited our analysis to promoters that had their TSS at least 15 nt upstream of the URR. Genes associated with transcripts harboring URRs with 100% AU had the highest TE (Fig. 5e). TE was significantly higher in the high-AU% group compared to the low-AU% group (Supplementary Table 2) in all growth conditions, suggesting that the AU content at the URR significantly impacts TE.

We further compared the position of the RBS relative to the translation initiation site and showed that genes with highest TE were those harboring RBSs 13 nt (from the 5′ end) upstream of the translation initiation site (Fig. 5f, Supplementary Data 2). In addition, we found that the most conserved RBS motifs tend to be at an optimum distance from the translation initiation site (Supplementary Fig. 7). For the coding region of mRNA, we analyzed the effect of codon usage on TE, using the codon adaptation index (CAI)[42]. Although the high-TE set had significantly more optimized codons (Mann−Whitney $U$ test, $P_{CAI} = 1.9e^{-07}$, Supplementary Fig. 8), the difference in the average AU content of the coding region was substantially more significant with the high-TE set having significantly lower AU content (Mann−Whitney $U$ test, $P_{AU\%} = 8.3e^{-60}$; Supplementary Fig. 8), suggesting that secondary structure strongly affects TE.

**High expression of optimized mRNAs maximizes TE**. Thus far, our results hint towards prioritization of subsystems involved in carbon and energy metabolism by differentially increasing their TE in response to available resources (Fig. 2). The mRNA features carried at the 5′UTR and at the coding region (Fig. 5) can both influence translation[43–45]. We currently lack insight into how static mRNA features in bacteria influence TE in a condition-specific manner. To address the link between mRNA level, mRNA features, and TE, we first determined the maximum TE ($TE_{max}$) for each gene across the three growth conditions tested and used it as a measure of the maximum affinity of the mRNA towards the ribosome. We split genes into quartile groups based on to their $TE_{max}$ values (Supplementary Fig. 9): high (75–100%), medium-high (50–75%), medium-low (25–50%), and low (0–25%). Quartiles were determined after retaining nonextreme TEs (0.01 < TE < 5) and mRNA FPKM > 10. We compared the optimization level of the mRNA features across all quartiles

(asterisks in Supplementary Fig. 9). Apart from RBS distance from start codon feature, which had only one "not significant" quartile, all quartiles in other features had significantly different TE from the high-TE quartile, confirming our previous findings that TE is highly influenced by these features.

Comparing translation to enzyme kinetics in which substrate competition is dependent on the concentration and the affinity of each substrate, we considered RNA-seq (FPKM) to represent the concentration of the substrate and $TE_{max}$ to represent the affinity. Since TE highly depends on the mRNA features (Supplementary Fig. 9), we expect that as the concentration of mRNA increases, those molecules with high affinity (higher $TE_{max}$) would attract more of the limited number of ribosomes and hence outcompete those with a lower affinity (lower $TE_{max}$). By plotting the TE quartiles versus RNA-seq (FPKM) categorized in ten groups, we observed that as the transcription level of highly optimized mRNA increases, the affinity (TE) also significantly increases (Fig. 6a, violet regression line, [$P$ (Wald) < $2e^{-16}$]). In contrast, for the low-TE group as mRNA level increases the TE significantly decreases (Fig. 6a, blue regression line, [$P$ (Wald) = $0.2e^{-06}$]). The high- and low-TE slopes are significantly different ($P$ (Turkey HSD) < 0.01). Our analysis suggests that mRNA molecules with high $TE_{max}$ are more competitive substrates for the ribosome. The increased competitive capacity becomes apparent at higher levels of transcription at which the number of ribosomes per mRNA will increase. At the same time mRNAs with suboptimized mRNA will show decreased TE because the number of ribosomes per mRNA will drop. Therefore, mRNA molecules with optimized features are prioritized for translation when their mRNA levels are increased.

**Distinct functions are enriched with optimized mRNAs**. We show that such mRNA molecules are significantly enriched ($P$ (Fisher exact test) < 0.05) in RAST categories involved in carbon and energy (respiration in Fig. 6b) metabolism. mRNA molecules with less optimized features are significantly enriched in housekeeping genes and mRNA molecules with least optimized features are enriched in RAST categories involved in dormancy and transcriptional regulation (Fig. 6b, Supplementary Table 3).

Overall, our results suggest that genes involved in carbon and energy metabolism are primed to be promptly translated in response to substrate availability. Optimized features of mRNA molecules increase their relative TE allowing them to outcompete less optimized molecules which support basic functions like cell maintenance.

## Discussion
Here, we carried out a multi-omics approach to study the translational control underlying important carbon and energy metabolism in the model acetogen *C. ljungdahlii*. RNA-seq and Ribo-seq data were combined from identical samples to ensure high robustness. We found that a sizable number of genes had TEs markedly above or below the average in all growth conditions, implying strong translational regulation. By using RAST functional enrichment, we demonstrated that carbon and energy subsystems are highly regulated at the translational level. These subsystems are enriched for highly optimized mRNA molecules that allow significantly higher TE and fast translational response. Whereas subsystems involved in cell maintenance are enriched for less optimized mRNA and slower response to fluctuations in mRNA levels.

Translation is an energy-expensive process and in energy-deprived niches it is essential to rationally assign translation resources to pathways that enable best fitness. AU content of the 5′UTR, the position and affinity of RBS, the codon usage and the

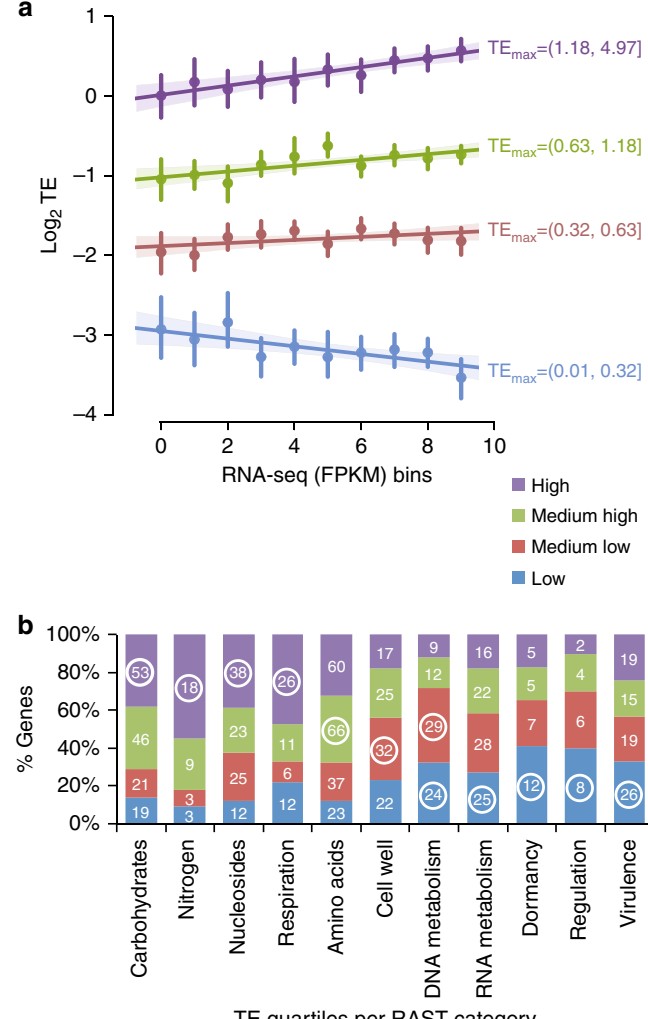

**Fig. 6** mRNA features and transcription levels determine the dynamics of differential TE. **a** Scatter plot between log$_2$ TE as four quartiles on the Y axis and RNA-seq (FPKM) values grouped into ten bins on the X axis. TE$_{max}$ cutoffs are shown next to each regression with the same color codes. The dots represent the mean of TE in each RNA-seq category. The vertical lines over each dot represents 95% confidence intervals of each bin. The linear regression lines are fitted with shadowed regions representing 95% confidence intervals. **b** Stacked barplot demonstrating the enrichment of genes in RAST categories within each TE$_{max}$ quartile (shown as percent) with the same color codes as in (**a**). The white numbers represent gene counts for each TE$_{max}$ quartile in a given RAST category. Significantly enriched quartiles (Fisher exact test) are indicated with circled gene count numbers

AU content of the coding region are features, which contribute to the recruitment of mRNA molecules to the ribosome and therefore, determine the efficiency with which a given mRNA is recruited and translated[36,46]. We measured these features in the 5′UTR and in the coding region that showed a clear effect on TE. By comparing enrichment of these features in highly translationally efficient and in highly translationally inefficient subsystems, we showed that AU content at the URR as well as at the coding region is a very important determinant of TE. In addition, RBS affinity to aSD and the distance of the RBS from the translation initiation site were also a critical determinant of translation.

We find that translation elongation factors (EFs) have higher TE than translation IFs (Supplementary Fig. 10a). EFs also have relatively higher translation compared to IFs (Supplementary Fig. 10b). Notably, EF-Tu, EF-G, and EF-P are present in two paralogs each. EF-Tu paralogs are 100% identical, whereas ET-G and EF-P paralogs are 27% and 24% identical, respectively. IF-3 is also present in two paralogs, which are 54% identical. This is consistent with the notion that mRNA translation, in bacteria as well as in eukaryotes, is highly regulated at the initiation stage and that the relatively low abundance of IFs favors the translation of mRNAs with most optimized 5′UTR at highest efficiency. The higher abundance of the EFs is likely to ensure efficient elongation after passing the initiation bottleneck. Thus, selective increase in TE for highly optimized mRNA could be further driven by the differential increase of EFs abundance relative to IFs. Similar discrepancies between the abundance of IFs and EFs have been reported in *E. coli*[47], suggesting that this mechanism could be common in bacteria. However, additional experiments are required to prove this hypothesis.

Further, we find that ribosomal proteins S6, S18, and S20 are significantly translationally upregulated in heterotrophic growth. S6 was demonstrated to form a complex with S18 and the S6:S18 heterodimer binds to specific motifs in their corresponding mRNAs and autoregulate their translation[48]. The TE of S18 is one third lower in heterotrophic condition (where it is most highly translated) compared to its value in the two autotrophic conditions, suggests negative auto-regulation by the S6:S18 complex (TE = 0.6 in heterotrophic compared to 0.9 in autotrophic conditions). S6 TE is only 0.1 and is stable in all conditions. In contrast, there are no significant translationally upregulated ribosomal proteins in the two autotrophic growth conditions. It is not clear why these ribosomal genes are differentially translated in heterotrophic growth. However, there are many lines of evidence demonstrating that specialized heterogenic ribosomes could translate a subgroup of mRNA molecules[49,50].

CsrA is a global translation factor that regulates the translation of at least 720 transcripts in *E. coli*[51]. It is found in most bacterial species and has also been shown to interact with the 5′UTR region by directly interacting with the RBS, effectively controlling translational initiation and mRNA stability[29]. It activates the translation of the glycolysis genes in *E. coli* and represses biofilm formation, but it has no effect on the pentose phosphate pathway genes[52]. A transcriptomics study on the *csrA* mutant has been reported in *Clostridium acytobutylicum*[53], which demonstrated that the transcription of some central carbon metabolism genes were affected, but since CsrA is a translational regulator, it is difficult to deduce concrete conclusions without ribosome profiling or proteomics analysis. A CsrA ortholog also exists in the *C. ljungdahlii* genome (*Clju_c09540*). Two small RNA molecules, CsrB and CsrC, antagonize the binding activity of CsrA in *E. coli*[54]; however, no corresponding homologs are detectable in the *C. ljungdahlii* genome. The *csrA* gene in *C. ljungdahlii* is differentially upregulated at the transcriptional level under autotrophic growth, but not at the translational level (see Supplementary Data 2 for gene expression details). Currently it is difficult to determine whether CsrA regulates the translation of the glycolysis genes, especially without the *csrA* deletion mutant and without identifying CsrB and CsrC, but it would be interesting to decipher its function in future studies, since it could potentially regulate genes in major fermentation pathways.

Besides mRNA features and translation factors, translation could be slowed by the formation of small secondary structure elements, especially small α-helical domains of the nascent peptide within the exit tunnel[55]. Such peptides often destined for

insertion into the membrane[56] and the increased ribosome pausing is required for proper folding[57]. We compared the TE of 782 membrane proteins (predicted by TM HMM[58]) with TE of 3374 cytoplasmic proteins. Consistent with previous studies[59,60] that showed significant slowdown in the translation of membrane proteins, *C. ljungdahlii* membrane proteins have significantly lower TE than cytoplasmic proteins in all growth conditions (Mann−Whitney *U* test, $P < 0.01$, Supplementary Fig. 11).

We provided examples of strong translational control in energy conservation pathways. We showed for the first time that the Rnf complex is translationally repressed under heterotrophic growth, a condition where the Rnf complex was shown to be dispensable[3]. The stoichiometry of the Rnf complex is still undetermined and the severe repression of the RnfC translation under heterotrophic growth raises the possibility of differential Rnf stoichiometry under autotrophic and heterotrophic growth. The formate dehydrogenase and the hydrogenase complexes showed no apparent difference in their TE in all growth conditions; the hydrogenase complex on average has higher TE than the formate dehydrogenase in all growth conditions. The ATPase genes were translationally inefficient regardless of the growth condition, implying strong translational stability that is independent of growth conditions.

Functional enrichment is a powerful tool that facilitates the understanding of resource allocation in different growth conditions. We carried out flux balance analysis (FBA) using a genome-scale metabolic model for *C. ljungdahlii* (iHN637)[2] to compare FBA results with different experimental measurements (**see** Supplementary methods). As expected, we did not observe any correlation at the gene level, but we found good correlation when we compare RNA-seq (Pearson's $R = 0.73$) and Ribo-seq (Pearson's $R = 0.70$) with fluxes at the subsystem level (Supplementary Fig. 12a and 12b; Supplementary Data 4). However, the correlation was only 0.15 when comparing fluxes with TE at the subsystem level (Supplementary Fig. 12c) and fluxes at the gene level were not correlated with TE (Supplementary Fig. 12d). Our experimental results could thus be a useful tool to validate and improve the predictability of metabolic models.

We took a systems-level approach to understand the overall TE and how it is linked to transcription and *cis*-acting regulatory sequences at the 5′UTR. The rate by which TE increases in response to increased mRNA levels depends mainly on the 5′UTR region. Highly optimized 5′UTRs are significantly enriched in growth condition-specific carbon and energy metabolic pathways, whereas suboptimal 5′UTRs are enriched in housekeeping genes. This prudent assignment of optimized 5′UTRs to carbon and energy pathways ensures faster translational response of urgently required pathways, which is vital when scarce resources are transiently available. By the same token, the assignment of suboptimal 5′UTRs to housekeeping genes (lower TE) ensures stable translation as well as the use of minimal resources to sustain maximal growth. Differential TE of metabolic pathways and genes has been reported previously for both eukaryotes[61] and prokaryotes[62]. The effect of the 5′UTR length and secondary structure on TE and their role in regulation of secondary metabolite translation has been reported in *S. coelicolor*[18]. However, the effect of 5′UTR and RNA-expression levels on TE and how this strategy is used to allocate resources efficiently was previously unknown. Here, we illustrate how the interplay between defined features in the 5′UTR and RNA transcription levels determines condition-specific TE of metabolic pathways. Moreover, we show how *C. ljungdahlii* modulates the TE levels for metabolic pathways in a growth condition-dependent manner. Our work unravels how acetogens utilize the differential TE mechanism to use carbon and energy resources optimally to thrive at the thermodynamic energy limit of life.

We propose that pathways involved in carbon and energy metabolism are specifically controlled through optimizing the TE, allowing for dynamic resource allocation. The findings have broad implications on how microorganisms control and optimize their metabolic networks. The results provide a new framework for metabolic regulation in this model acetogen that can readily be extrapolated to other industrially important microbes. Unraveling of regulatory mechanisms lays the foundation for advanced strain design and engineering efforts.

## Methods

**Growth conditions**. For heterotrophic growth, *C. ljungdahlii* (ATCC 55383) was grown anaerobically without shaking to mid log phase in 125 ml serum bottles containing 100 ml of PETC medium (ATCC medium 1754) supplied with 28 mM fructose at 37 °C. For low-fructose TSS-seq analysis *C. ljungdahlii* was grown in PETC medium supplied with 4 mM fructose. Autotrophic growth was carried out in 1 l bottles containing 300 ml PETC medium and pressurized with either 1.8 bar of CO gas or H2:CO2 (1:4 vol/vol) gas mix. Cultures for further analysis were collected at mid log phase.

**Ribo-seq library preparation**. All experiments were performed using two biological replicates. Ribo-seq was performed as described previously[63] with minor modifications. In brief, cell pellets were collected by centrifugation at room temperature for 5 min at 5000 rcf (relative centrifugal force). The growth medium was removed and cell pellets were immediately snap frozen in liquid nitrogen then kept at −80 °C. Cell lysates were prepared by grinding the pellets in liquid nitrogen together with 350–400 μl lysis buffer containing 50 μg/ml thiamphenicol. The lysates were cleared by centrifugation at 4 °C. In order to enable direct comparison between Ribo-seq and RNA-seq, 100 μl of cleared cell lysates were stored at −80 °C for RNA-seq library preparation with 500 μl of Trizol reagent (Thermo Fisher Scientific) for stabilization. The remaining lysates were treated with MNase to remove free RNA and DNA. The reaction was stopped by adding EGTA as described in the original protocol[63]. Ribosome-bound mRNA footprints (RBF) were harvested by filtering through Sephacryl S400 MicroSpin Columns (GE Healthcare, Piscataway, NJ, USA) followed by the protocol of small RNA isolation as part of the miRNeasy MINI Kit (Qiagen). The ribosomal RNA was removed from RBF by Ribo-Zero rRNA Removal kit for bacteria (Illumina). Sequencing libraries were made using the NEBNext® Small RNA Library Prep Set for Illumina kit (New England Biolabs).

**RNA-seq library preparation**. The lysates stored during Ribo-seq library preparation were brought to room temperature. 140 μl of chloroform was added to each tube, vortex-mixed and centrifuged at 4 °C for 10 min. The aqueous fraction was isolated and total RNA was extracted using the RNeasy mini kit (Qiagen) followed as recommended by the manufacturer's instruction with minor modifications. In brief, the volume was brought to 900 μl using RLT buffer and 600 μl of 95% ethanol was added and mixed in order to bind the RNA. Eighty microliters of RNase-free DNase mix (8 μl DNase I (Roche) of 10 U/μl in 80 μl total volume) was added to the RNeasy MinElute columns. The ribosomal RNA (rRNA) was depleted using the Ribo-Zero rRNA Removal kit for bacteria (Illumina). rRNA-depleted RNA was used for RNA sequencing (RNA-seq) and TSS-seq. Strand-specific RNA-seq libraries were prepared using the Stranded mRNA-seq Kit (Kapa Biosystems) as per the manufacturer's instruction.

**TSS-seq library preparation**. Strand-specific TSS-seq libraries were prepared using the 5′RNA-seq protocol[64] with minor modifications. The rRNA-depleted RNA (see above) was split into two samples for two different libraries: the library of the whole transcriptome and the library of the processed transcriptome, respectively. RNA 5′-polyphosphatase (RPP) was treated only for the library of whole transcriptome. The 5′-RNA adaptor (ACACUCUUUCCCUACACGACGCU-CUUCCGAUCU) was ligated to both RPP-treated and untreated samples. Then, cDNA was synthesized using the adaptor with random nonamer for reverse transcription compatible with TruSeq primers (GTGACTGGAGTTCA-GACGTGTGCTCTTCCGATCTNNNNNNNNN). TruSeq i7 indexed adaptors were used for amplification to enable sample multiplexing.

**High-throughput sequencing**. The libraries from all experiments were sequenced in either the Illumina HiSeq™ 4000 or MiSeq™ instruments at UCSD IGM genomics center. The Ribo-seq libraries were sequenced using 50 bp-cycle kits, and TSS-seq and RNA-seq libraries were sequenced using 100 bp-cycle kits.

**Data processing**. The adapter sequence was trimmed, and low-quality reads were removed. The reads were aligned to the *C. ljungdahlii* genome (accession no. NC_014328) using bowtie2 [65] with one mismatch allowed. Samtools[66] (version 1.4.1) was used to filter out low-quality alignments ($q < 10$). The number of reads per gene was obtained using Subread package-featureCounts (version 1.5.0-p1)[67].

RPKM, FKPM and correlation values of biological replicates were calculated using Python (2.7) in-house scripts in jupyter notebook format. We used in-house Perl scripts to generate wig files for genome-wide track viewing. Relative expression ratios were obtained for three different paired samples: (i) fructose versus CO, (ii) fructose versus $H_2$:$CO_2$, and (iii) CO versus $H_2$:$CO_2$. Differentially transcribed and translated genes were identified using DESeq2[20] package in R (3.3.1), with the following parameters settings: adjusted $P$ value < 0.001, log2-fold change ≥ 1. Correlation values between RNA-seq and Ribo-seq were calculated using the mean value of the biological replicates for each experiment. Translational efficiency was obtained by the ratio between translation and transcription levels per gene. Differential-efficiently translated genes were obtained following the criteria: (i) genes translated in both biological replicates (normalized counts ≠ 0); (ii) genes transcribed in both biological replicates (normalized counts ≠ 0); (iii) high-differential TE genes > 80th percentile; (iv) low-differential TE genes < 20th percentile.

**RNA-seq and Ribo-seq data analyses.** The *C. ljungdahlii* genome (accession no. NC_014328) was annotated in the RAST database in order to systematically organize genes into categories, subcategories, and subsystems. DESeq2 package in R was used to find significant translation and transcription of individual genes and RAST subsystems[20]. RPKM and FPKM values for translation and transcription, respectively, were summed for genes per each RAST subsystem. We calculated the total number of sequencing reads that were aligned to each RAST subsystem, regardless of whether genes in a given subsystem are differentially regulated or not. We then calculated the RPKM/FPKM for each subsystem and reported the values of the top 20 subsystems that were significantly differentially transcribed/translated according to DESeq2. We assumed that the amount of resources required for transcription/translation will be similar whether it is assigned to one gene or a group of genes in a given RAST subsystem.

For each growth condition, subsystems were normalized as a percent ratio of the total translation or transcription for subsystems annotated by RAST (i.e. genes that are not in RAST categories were not considered). To directly compare translation and transcription across growth conditions, the normalized percent values described above, were summed for all growth conditions and normalized to percent. TE was calculated by dividing subsystem percent Ribo-seq by subsystem percent RNA-seq.

Translation at the subsystem level was considered significantly differentially efficient in heterotrophic growth if three conditions were met. First, the fructose subsystem translation has adjusted $P$ value < 0.01 when compared to the translation of the CO and the $H_2$:$CO_2$ subsystems, but CO subsystem translation versus $H_2$:$CO_2$ was insignificant with adjusted $P$ value > 0.01. Second, fructose subsystem translation >50th percentile. Third, TE of fructose > mean of CO TE and the mean of $H_2$:$CO_2$ TE. In autotrophic growth, subsystems were considered significantly differentially efficient if three conditions were met. First, both CO and $H_2$:$CO_2$ subsystem translation levels have adjusted $P$ values < 0.01. Second, both CO and $H_2$:$CO_2$ subsystems are above their 50th percentiles. Third, both CO and $H_2$:$CO_2$ subsystem translation levels > the mean of fructose subsystem translation level.

**TSS identification and categorization.** To maximize TSS detection, we carried out TSS-seq in four growth conditions. These consisted of two heterotrophic conditions (fructose and low-fructose (see above)), and two autotrophic conditions (CO and $H_2$:$CO_2$). For each growth condition, RNA 5′-polyphosphatase (RPP)-treated and untreated samples were sequenced in duplicates. Genomic positions at the 5′-end of uniquely aligned TSS-seq reads were considered to be potential TSSs such that TSS is uniquely associated with RPP-treated samples. At this point we had eight pairs of datasets from four conditions. A TSS was scored if a sequencing read was present in the RPP-treated sample, but absent in the corresponding untreated sample. Thereby, we scored TSS from one to eight, with the former being found in only one dataset and the latter being found in all datasets and in all conditions. Finally, the potential TSSs were visually investigated and compared to corresponding RNA-expression levels in order to have stronger confidence. Identified TSS positions were categorized by genomic location. TSSs located 250 bp upstream of start codon to 50 bp downstream of the start codon were annotated as Primary TSSs (P). TSSs located within coding regions at 50 bp downstream of the start codon to end of the stop codon were annotated as Internal TSSs (I). TSSs located antisense relative to any open reading frame were annotated as Antisense TSSs (A). TSSs that do not fall into any of the categories above were annotated as Orphan TSSs (O). Operon coverage was calculated by multiplying the mean operonic gene content (1.67) by the total number primary TSS (1245) divided by the total number of genes (4182). Operons were predicted using the DOOR tool[68].

**5′UTR analysis.** We used a method described by Li et al.[40] with some modifications to detect putative ribosome-binding sequences (Shine-Dalgarno). A fasta file containing 25 nt upstream of the translation initiation site of each gene was generated. RNAplot[69] was used to calculate the 25-nt regions affinity towards anti-SD sequence (AAGGAGGU) with 10-nt walking windows that scan the entire 25 nt. The window with the lowest −ΔG score was considered to be the one harboring the RBS. This approach was validated by calculating the distance of the putative RBS from start codon and ensuring that the 5′-RBS mean distance from the start codon is 13 (3′ of RBS is 8 bp upstream of RBS). Further analyses were carried out using an in-house Python script. The putative RBS binding motifs for each category was generated using MEME[70]. AT content was calculated in percent for the 15 bp upstream of each putative RBS (URR), such that the TSS is upstream of the 5′-end of the URR.

**Code availability.** Computer code will be provided upon request from corresponding author.

## Data availability

Sequencing files were deposited to NCBI SRA site under the bioproject IDs PRJNA418190 and PRJNA395392. Proteomics data were deposited to ProteomeXchange and MassIVE under the accession codes PXD009162 and MSV000082141, respectively.

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

## Acknowledgements

We are grateful to Richard Szubin, Cameron Martino, and Jana Tarasova for their technical support. We would also like to thank Dr. Haythem Latif, Dr. Elizabeth Brunk, and Justin Tan for their helpful advice and discussions on ribosome profiling. This study is based on work supported by the U.S. Department of Energy, Office of Science, Office of Biological & Environmental Research Awards DE-SC0012658 and DE-SC0012586. The Graduate Training in Cellular and Molecular Pharmacology Training Grant, NIH T32 GM007752, supported J.M.W.

## Author contributions

Conceptualization: K.Z., M.M.A.-B. and J.-N.K.; Experimental work: M.M.A.-B., J.-N.K., J.M.W. and D.J.G.; Omics data analysis: M.M.A.-B., J.K., L.S.Z., C.Z. and B.P.K.; Writing original draft: M.M.A.-B., J.-N.K. and K.Z.; Discussion: M.M.A.-B. and K.Z.; Funding acquisition, resources and supervision: K.Z.

## Additional information

**Competing interests:** The authors declare no competing interests.

