## [Peer Review File · Nature Communications]

Reviewers' comments:

Reviewer #1 (Remarks to the Author):

Al-Bassam et al. present a contribution, which claims that translation efficiency in the acetogen *C. ljungdahlii* is differentially elevated under specific growth conditions. This is effected by coding and untranslated regions of mRNA. Such a finding would be highly interesting, as acetogens emerge as potent microbial production platforms, converting cheap waste and greenhouse gases such as CO and CO₂ into value-added bulk and speciality chemicals.

However, there are a number of shortcomings with this contribution that need to be addressed:

1. The authors use the established term "translational efficiency". However, they define it as dividing the translational level by the transcriptional level (of a given gene) (l. 75/76). This is in sharp contrast to the official definition as protein per mRNA per hour (Schwanhäusser et al., Nature 473, 337-342, 2011). Thus, Al-Bassam et al. only used RNA-based techniques, proteins were never measured, they introduce a value consisting of amounts, not a rate! I really doubt that one can make conclusions on translation, when products of this process (i.e. proteins) have never been measured. The authors base their conclusions solely on ribosome-bound mRNA compared to free mRNA. This is in my opinion a major pitfall of this study.

What could be done? If really verified, the results of this study are really interesting, so they do have merit. However, verification is essential. Ideally, the study would be complemented by a proteome analysis, which definitely would prove or disprove the authors' conclusions. Well, I am of course aware of the fact that this would be a major undertaking and a publication in itself.

However, the authors could perform additional experiments on some selected proteins (e.g. Rnf complex) and see, if the data collected verify their conclusions.

2. Especially the data presented on rnf genes are of severe concern. The authors conclude that the Rnf complex is highly translationally repressed, especially for rnfC and rseC (l. 167, Fig. 4b). As evident from this figure, rnfC is transcribed together with rnfD and rnfG, which are not similarly effected by the different growth conditions. That would be really strange, and the authors do not provide a convincing explanation for this phenomenon. However, looking into the genome sequence of *C. ljungdahlii*, it becomes obvious that there is another gene with annotated similarity to rnfC. If the authors did not differentiate between these two genes during their analysis and both are differently expressed, this might explain this result. However, this clearly would render the result erroneous. In any case, the reader would like to know, how the authors differentiated between very similar genes!

3. In Fig. 3, several metabolic reactions are depicted and changes of their "translation ratio" are listed. However, some reactions are simply mentioned as with insignificant changes. This is really surprising for single genes in the middle of a pathway (e.g. W6, methylene-THF dehydrogenase), described as "mostly differentially regulated under autotrophic growth" (l. 148/149).

4. Possibly directly related is the statement of the authors that they only performed "two biological replicates" (l. 395). The accepted scientific standard is the use of triplicates to minimize statistical errors. Where there differences in measurements of reaction W6?

5. Several minor criticisms concern the presentation of the contribution. Bacterial names should always be in italics (e.g. l. 25), subscripts should be followed with consisting formatting (e.g. l. 23, 29), references should be separated by commas (e.g. l. 31), abbreviations need to be explained (e.g. l. 65, 447).

Reviewer #2 (Remarks to the Author):

SUMMARY:

Manuscript titled "Optimization of Carbon and Energy Utilization Through Differential Translational Efficiency" by Al-Bassam et al., investigates translational regulation under autotrophic and heterotrophic conditions in the model acetogen *Clostridium ljungdahlii*. Authors use RNA-seq, Ribo-seq and TSS-seq in order to acquire measurements for transcription, translation and transcription

start site information. They determine translational efficiency (TE) under three conditions reflecting two major growth modes and associate this information with RAST annotated functional information in order to identify differential translational regulation for different functional categories. They also use TSS-seq to define 5-UTR regions and connect these molecular features to differential TE of major functional categories.

MAJOR COMMENTS:

It was shown before that differences in translational efficiency between transcripts appear to play a prominent role in stress responses or during environmental alterations. Most of the initial studies focused on determination of transcript abundances through RNA-seq and comparing it to the protein abundances through quantitative proteomics to determine TE. Recently, development of ribosomal profiling through sequencing (Ribo-Seq) allowed to determine actively translated transcripts and compare it to RNA-seq based transcriptomic measurements. For example, a recent study in *Streptomyces coelicolor* A3(2) determined the transcriptional and translational landscape, and TSSs for transcribed sequences and associated this information with primary and secondary metabolism (Jeong et al., 2016, Nature Communications).

In this manuscript, Al-Bassam et al., take the similar approach in *C. ljungdahlii* to measure translational efficiency under different growth conditions and find that high-TE is associated with carbon and energy metabolism related genes. These claims are well-supported by the comprehensive sets of data. This is an important step forward to understand the biology of this model acetogen and provided datasets would be a great reference for future studies. However, novelty of the findings and the uniqueness of the omics approach may not be of general interest to Nature Communications audience and may benefit from publication in a more specialized journal.

SPECIFIC COMMENTS:

1. There are some critical mistakes in the manuscript that makes it difficult to understand the results and manuscript requires a major proofreading. Some of these issues are listed below.
2. Discussion is merely a repetition of the results and does not include the overall impact of the current work for the field, connections to other studies and scientific significance. I would strongly recommend expanding on the interpretation of the results to highlight the significance and implications of the research and what needs to be done next. Also, it can benefit from the discussion of the potential technical and data analysis issues and challenges with Ribosomal profiling, TSS-seq and combining multi-omics data sets.
3. Reference numbers for single digit references are not formatted correctly. e.g 123 instead of 1,2,3.
4. Figure numbers are missing in the figures.
5. Page 1, L23, 29: H₂ or CO₂, "2" must be subscripted
6. Page 1, L25: *Clostridium ljungdahlii* must be italic
7. Page 2, L56: "...codon adaptation control TE and are crucial factors.." "and" should be removed.
8. Page 2, L65: RPKM and FPKM normalization is referred to Supplementary Figure S1 but there is no information in that figure about these. Also, it is not clear how percent efficiency was calculated in Supplementary Figure S1.
9. Page 3, L 75-77: "We calculated the TE of each gene by dividing the translational level by the transcriptional level and noticed significant discrepancy in TE among different genes (Fig. 1b)." Figure 1b only shows a selected region on the genome. It would be nice to have a high-level statistic on how common is this trend and also labeling for what genes are specifically highlighted in this figure would be nice.

10. In Figure 2, it is not clear how many genes are in a given subsystem. Is it possible that enrichment can be biased by few genes in a given subsystem with very high RPKM or FPKM. They already see an example of this effect with "sporulation cluster (H4)"

11. Page 7, L206: "...we compared RNA-seq and Ribo-seq data of genes with low-TE (<20th percentile) or low-TE (>80th percentile)."

Do you mean or "high-TE (>80th percentile)" instead of or "low-TE (>80th percentile)"?

12. Page 7, L241-243: "The difference was highly significant (Supplementary Table 2) between the two groups under all three conditions, suggesting that RBS affinity towards the initiating ribosomes is a key determinant for TE."

Should "Supplementary Table 2" be "Supplementary Table 3"?

13. Page 9, L274-275: "Features that promote low-TE are enriched in differentially transcribed genes involved in condition-specific carbon and energy metabolic pathways".

Do you mean "Features that promote high-TE are enriched in differentially transcribed genes involved in condition-specific carbon and energy metabolic pathways"

14. Page 9, L285-287: "...carbon and energy metabolism enriched in genes that are differentially transcribed and show a low-TE, but not enriched in differentially transcribed and low-TE genes."

Do you mean "...carbon and energy metabolism enriched in genes that are differentially transcribed and show a high-TE, but not enriched in differentially transcribed and low-TE genes."

15. Page 9, L290-292: "Group 1 represents our "test" group and consist of low-TE genes (>80th percentile) for both autotrophic growth (blue and green dots in Fig. 6a, b)"

Do you mean "Group 1 represents our "test" group and consist of high-TE genes (>80th percentile) for both autotrophic growth (blue and green dots in Fig. 6a, b)"

16. On Page 9, authors calculate RAST-enrichment as the ratio of the genes in group 1 or 2 to total number of genes in a given subsystem and rank the ratios. It is not totally clear to me why they would calculate enrichment this way? I think the more appropriate way would be to calculate hypergeometric p-values, after correction for each subsystems and report significant enrichments.

17. Page 10, L315-234: This paragraph and Table 1 mainly report results related to UTR features and their impact on TE. It would be more appropriate to move this paragraph to the end of the results section where authors discuss the impact of these features (Pages 6-8).

18. Methods section, Data Processing: "RPKM, FKPM and correlation values of biological replicates were calculated using statistical software R (version 3.3.1)."

This is a very generic description and calculation of RPKM and FPKM are central to this paper.

Therefore, authors should describe whether calculations are performed by using a specific R package or custom scripts in which case they should provide these custom scripts.

19. Figure Legends, Figure 1: "RNA-seq and Ribo-seq profiles were normalized in RPM."

What is RPM? Reads per million?

20. Figure Legends, Figure 2: "Differential translation and differential TE of subsystems in fructose"

Do you mean "Differential transcription and differential TE of subsystems in fructose"

Figure 5a, condition names are missing. What are the units?

Reviewer #3 (Remarks to the Author):

Article Summary:

Ribosome profiling has in recent years been used to show that there is prevalent translational regulation of gene expression under various conditions. The submitted study compares transcriptional and translational regulation of gene expression in *C. ljungdahlii*, a model acetogen organism, grown under different growth media conditions. To do this they used a combination of RNA-seq and Ribo-seq and TSS-seq (to define transcriptional start sites genome-wide). The most interesting results of the study, although perhaps not surprising given the studies on other organisms, performed in the last couple of years, is that translational control is most prevalent in those metabolic pathways that are the most useful for growth under given conditions – heterotrophic vs autotrophic growth. The authors try to explain the molecular mechanisms of translational control, which is where, in my opinion the study lacks clarity and explanatory power. All in all, this is an interesting study not only for the acetogen research community, but also for the wider community, interested in the interplay between transcription and translation. If the interpretational part of the study is improved, I would recommend it for publication.

Major issues:

1) How novel is the study? While this is, to my knowledge, the first time that translational regulation has been studied genome-wide in an acetogen, there have been other studies which did the same for other prokaryotes and eukaryotes. These studies have also found major translational regulation. I would like the authors to do a more thorough comparison with previous studies and evaluate whether they actually found something new or are they presenting an old message on one more organism?

2) I find the last two sections of the results very difficult to read and understand, partly because of several typos (see below and under minor issues). In my opinion, in these sections the authors confuse two similar, but different questions. One is, what are the features of mRNA molecules that determine whether they will be translated efficiently or not. The authors test the CAI, AU-richness of 5'UTR, strength of the RBS motif and find these all influence translational efficiency. However, these features are permanent, therefore they cannot explain why translational efficiency changes when bacteria are exposed to different growth conditions. Since different genes are translated efficiently in heterotrophic and autotrophic growth, one should not be looking at similarities of all high-TE genes, but at the differences between these groups – but the authors have not done so. Instead they put forward a hypothesis that (lines 282-287) “if features in the UTRs and in the coding regions have significant influence on TE in a mRNA-dependent manner, we expect to find subsystems related to carbon and energy metabolism enriched in genes that are differentially transcribed and show a low-TE, but not enriched in differentially transcribed and low-TE genes. “ This is an extremely confusing sentence. Firstly, why would there be transcriptional regulation of translation efficiency? In Figure 1, Ribo-seq vs RNA-seq correlation is shown – if there was transcriptional dependent TE changes, would not an even better correlation would have to be there for TE vs RNA-seq? Secondly, in both groups low-TE genes are used (see also lines 291 vs 293), I guess one of these should be changed to high-TE?

In the end they find an enrichment of carbon metabolism genes in the differentially expressed groups with high TE – but this should be expected already from Figure 2, where the almost same enrichment is clearly there. It also does not show that TE is in any way transcriptionally dependent, just that an increase in TE correlates with differential transcription in carbon metabolism. I just don't see how any of these supports the claim that translation of these genes is affected by fluctuations in transcription (lines 312-313).

3) The only time the authors at least implicitly recognize that there needs to be a molecular mechanism behind the differential translation under different conditions, is in lines 251-260, where they mention the RPS1 (30 ribosome protein S1) gene. If this gene is responsible for differential translation, then it would be interesting to find out whether it was differentially expressed in the different conditions, actually it would be interesting to find out whether any genes known to be involved in translation initiation are differentially expressed. It is possible that the molecular mechanism could be found among these genes, as was the case in other studies – see e.g. Hsieh AC, Nature 485; Thoreen CC, Nature 485 for mTOR regulated translation in mammals, which found 4E-BP, eIF4G1 and eIF4E to be the mediators of differential translation.

Minor issues:

- 4) So, one of the more interesting results of the manuscript is the selected increase in translational efficiency for metabolic genes connected to growth under heterotrophic or phototrophic conditions. Since, a metabolic reconstruction is available for *C. ljungdhalii* (ref 2), it would be interesting to see how the obtained high-TE genes match the amplitude of metabolic fluxed under heterotrophic and autotrophic growth obtained from flux balance analysis.
- 5) The authors use RPKM and FPKM metrics for gene expression and also used them to compare between different samples (lines 158-160). It has been shown that the TPM metric is better in this case.
- 6) Figure 2, Supplementary Figure 3. Using the sizes and colors of circles is an interesting way of presenting a lot of information in a single figure, but this means that it needs to be explained perfectly. My interpretation is that one should look at color in columns within an experimental conditions (dark red would mean that there are more transcripts, blue less), while sizes in rows and compare between conditions. This should be made even clearer in the figure legends. Also, not sure whether it's necessary to change the colormap for the TE.
Finally, in caption of Figure 2, the term translationally induced is used, while in Methods (Lines 470-479), significantly differentially efficient (SDE) subsystems are defined. I assume SDE and translationally induced is the same, but it would be helpful if the same terms were used throughout the manuscript.
- 7) Referencing within the text does not have any commas, which in some cases makes it difficult to read which references are supposed to be referenced. Example on line 72 9810.
- 8) Line 47: „we carried out cognitive analysis“. I haven't come across this term yet, what exactly is cognitive analysis?
- 9) Lines: 116-117: „A2 and A4 had the most diff TE subsystems“. Maybe better - > „were“ the most diff TE subsystems?
- 10) Lines 234-235: „ we compared the composition of the -10 and -35 regions of the 5' UTR by analyzing 40nt upstream of the TSS“. Should read 40 nt upstream of the start codon, see also Figure 5b.
- 11) Lines 248-249: „whereas groups with low-TE exhibit an optimal RBS motif“. Should read group with high-TE.
- 12) Line 282: missing and
- 13) Lines 457, 458: „iii) high-differentially translated genes – 90th percentile, vi) low-differentially translated genes – 10th percentile“. In the text in lines 206-207 low-TE and high-TE are described as (><20, >80%) Why are different numbers used for the same thing? . Also, vi) should be iv).
- 14) While the whole section „features that promote low-TE „ is confusing, I find that especially the term low-TE is inappropriately used throughout the section (should be replaced by high-TE in several places).

Optimization of Carbon and Energy Utilization Through Differential Translational Efficiency

We would like to express our gratitude to the reviewers and the editor for their time in reviewing our paper and for their insightful comments and suggestions. We tried our best to address their concerns and amend the manuscript as guided by the reviewers' suggestions. We have carried out all additional experiments and analyses suggested by the reviewers and believe that the manuscript has improved substantially. We hope that the reviewers will find our responses and our revised manuscript sufficient and convincing.

Reviewer #1 (Remarks to the Author):

Al-Bassam et al. present a contribution, which claims that translation efficiency in the acetogen *C. ljungdahlii* is differentially elevated under specific growth conditions. This is effected by coding and untranslated regions of mRNA. Such a finding would be highly interesting, as acetogens emerge as potent microbial production platforms, converting cheap waste and greenhouse gases such as CO and CO₂ into value-added bulk and speciality chemicals. However, there are a number of shortcomings with this contribution that need to be addressed:

1. The authors use the established term "translational efficiency". However, they define it as dividing the translational level by the transcriptional level (of a given gene) (l. 75/76). This is in sharp contrast to the official definition as protein per mRNA per hour (Schwanhäusser et al., Nature 473, 337-342, 2011). Thus, al-Bassam et al. only used RNA-based techniques, proteins were never measured, they introduce a value consisting of amounts, not a rate! I really doubt that one can make conclusions on translation, when products of this process (i.e. proteins) have never been measured. The authors base their conclusions solely on ribosome-bound mRNA compared to free mRNA. This is in my opinion a major pitfall of this study.

Regarding the cited paper (Schwanhäusser et al., Nature 473, 337-342, 2011), the authors of that paper have only compared absolute mRNA levels (transcriptomics) with absolute protein levels, which they used to calculate translation rate. They have not defined the term "translational efficiency". We implemented the definition and calculation of translational efficiency from the pioneering paper of ribosome profiling¹. This definition and calculation used here is widely accepted and implemented in the literature^{2,3,4,5,6,7}.

What could be done? If really verified, the results of this study are really interesting, so they do have merit. However, verification is essential. Ideally, the study would be complemented by a proteome analysis, which definitely would prove or disprove the authors' conclusions. Well, I am of course aware of the fact that this would be a major undertaking and a publication in itself. However, the authors could perform additional experiments on some selected proteins (e.g. Rnf complex) and see, if the data collected verify their conclusions. Especially the data presented on

rnf genes are of severe concern. The authors conclude that the Rnf complex is highly translationally repressed, especially for rnfC and rseC (l. 167, Fig. 4b). As evident from this figure, rnfC is transcribed together with rnfD and rnfG, which are not similarly effected by the different growth conditions. That would be really strange, and the authors do not provide a convincing explanation for this phenomenon.

We understand the concern raised by the reviewer.

1. It is well known that proteomics data are not well-correlated with transcription and ribosome profiling data¹. This is largely due to limitations in proteomics analysis⁸ but moreover, the post-translational degradation has been shown to be a less significant factor in determining the final protein titer in the cell⁹. In *E. coli*, absolute protein copy numbers were shown to be highly correlated with Ribo-seq data¹⁰, suggesting that ribosome profiling could potentially be an accurate method for measuring protein abundance.
2. The expression level of proteins in the cell is subject to two opposing constraints namely, the requirement of their function, and the cost associated with their synthesis¹¹. With this logic in mind, we expect the translation of the Rnf complex genes to be highly increased in autotrophic growth compare to heterotrophic growth. In fact, it has been shown that the Rnf complex is not essential in heterotrophic growth, but a deletion mutant of *rnfAB* was unable to grow autotrophically¹².
3. Unfortunately, we do not have the capacity to perform absolute protein quantitation and we feel this is out of the scope of the paper. However, we have carried out shot-gun proteomics as reviewer #1 suggested. In particular, we carried out proteomic analysis for cells grown under heterotrophic (fructose) and autotrophic (H₂:CO₂) conditions, in biological duplicates. The data corroborates our ribosome profiling data. We have added the results as Figure 4e. We also added the detailed methods in the supplementary methods file and the normalized relative abundance values in Supplementary table 5.
4. *C. autoethanogenum* and *C. ljungdahlii* are two phylogenetically indistinguishable species as they have >98% overall genome similarity¹³. This was also confirmed by pan genome analysis¹⁴. Quantitative proteomics carried out in *Clostridium autoethanogenum*, showed that RnfC (CAETHG_3227), RnfD (CAETHG_3228), and RnfG (CAETHG_3229) are all highly upregulated under autotrophic growth¹⁵. These findings from *Clostridium autoethanogenum* are identical to what we found in our RNA-seq, Ribo-seq and proteomics in *C. ljungdahlii*.

However, looking into the genome sequence of *C. ljungdahlii*, it becomes obvious that there is another gene with annotated similarity to rnfC. If the authors did not differentiate between these two genes during their analysis and both are differently expressed, this might explain this result. However, this clearly would render the result erraneous. In any case, the reader would like to know, how the authors differentiated between very similar genes!

1. It is true that there is an RnfC paralogue (RnfC2). However, the id% at the nucleic acid level is fairly poor (45%, using Clustal Omega [<https://www.ebi.ac.uk/Tools/msa/clustalo/>]), suggesting that reads are very unlikely to align to both proteins.
2. In order to confirm this, we made three indexes to align reads to:

- a. Index1 has both *rnfC* and *rnfC2* sequences
 - b. Index2 has only *rnfC* sequence
 - c. Index3 has only *rnfC2* sequence
3. We aligned the H₂:CO₂ Ribo-seq data to all three indexes using Bowtie2. We chose this particular experiment because it has *rnfC* highly expressed and also because Ribo-seq has shorter reads (average ~ 25nt), which are more likely to align to both genes than RNA-seq which has much longer reads.
 4. We compared the percent ratio of reads aligned to index2/index1 with percent ratio of reads aligned to index3/index1. It is obvious that almost all reads have aligned to RnfC and not RnfC2.

Index	# of reads aligned	% reads aligned
rnfC	7302	99.958
rnfC2	3	0.042
rnfC+rnfC2	7305	100

3. In Fig. 3, several metabolic reactions are depicted and changes of their "translation ratio" are listed. However, some reactions are simply mentioned as with insignificant changes. This is really surprising for single genes in the middle of a pathway (e.g. W6, methylene-THF dehydrogenase), described as "mostly differentially regulated under autotrophic growth" (l. 148/149).

We are grateful to the reviewer for raising this point. It is an interesting point that will add to our discussion. Regarding the "translation ratio" term, we reported the widely accepted measurement of translation efficiency, which is in fact, as reviewer #1 suggests, a translation ratio. Nevertheless, since the term has been widely implemented in the literature, we believe it would be confusing for the reader if we change this widely established term in our paper.

We believe the correlation between the activity of protein and its translation (or its abundance in the cell) is not always necessarily linked. There are several reasons for this. First, the activity of some metabolic enzymes can be regulated allosterically. Methylene-THF reductase is such an enzyme and it has been shown that NADPH and adenosylmethionine allosterically regulate the activity of methylene-THF in a mutually antagonistic manner^{16,17}. Therefore, it is difficult to understand the regulatory dynamics underlying this gene/protein without further biochemical assays, which we believe are out of the scope of this paper, but would be important to investigate in a follow up study. Second, protein half-lives vary considerably between proteins^{18,19,20}. Third, we carried out flux balance analysis (FBA) using a genome-scale metabolic model for *C. ljungdahlii* (iHN637)²¹ to compare FBA results with different experimental measurements. We do not see any correlation at the gene level, but we can see good correlation when we compare RNA-seq and Ribo-seq with fluxes at subsystem level (obtained from RAST). This additional analysis has now been added into discussion and Supplementary Fig. 12, as requested by reviewer #3.

4. Possibly directly related is the statement of the authors that they only performed "two biological replicates" (l. 395). The accepted scientific standard is the use of triplicates to minimize statistical errors. Where there differences in measurements of reaction W6?

We agree with reviewer #1 that including more biological replicates will, in many cases, improve statistical robustness. Nevertheless, we believe that two biological replicates are still statistically robust given that the correlation between the two replicates is very high, i.e. biological samples have very low variance. In our case, all experiments showed high Pearson's correlations (Pearson's correlations for RNA-seq = 0.995, 0.991, and 0.989, respectively and Pearson's correlations for Ribo-seq = 0.995, 0.952, and 0.930, respectively, Supplementary Fig. 2). Our claim is supported by a vast number of research papers where RNA-seq was performed by two biological replicates^{2,20,22,23,24,25,26}. There is some minor variance in the fructose experiment, however the average of the two fructose Ribo-seq experiments is in agreement with insignificant change between the three conditions.

CO 1 (RPKM)	CO 2 (RPKM)	Fructose 1 (RPKM)	Fructose 2 (RPKM)	H ₂ :CO ₂ 1 (RPKM)	H ₂ :CO ₂ 2 (RPKM)
2344.6	2793.4	4179.1	2987.8	2799.8	2782.9

5. Several minor criticisms concern the presentation of the contribution. Bacterial names should always be in italics (e.g. l. 25), subscripts should be followed with consisting formatting (e.g. l. 23, 29), references should be separated by commas (e.g. l. 31), abbreviations need to be explained (e.g. l. 65, 447).

We are thankful to the reviewer for pointing out typos and other mistakes. We have now corrected all the mistakes.

Reviewer #2 (Remarks to the Author):

SUMMARY:

Manuscript titled "Optimization of Carbon and Energy Utilization Through Differential Translational Efficiency" by Al-Bassam et al., investigates translational regulation under autotrophic and heterotrophic conditions in the model acetogen *Clostridium ljungdahlii*. Authors use RNA-seq, Ribo-seq and TSS-seq in order to acquire measurements for transcription, translation and transcription start site information. They determine translational efficiency (TE) under three conditions reflecting two major growth modes and associate this information with RAST annotated functional information in order to identify differential translational regulation for different functional categories. They also use TSS-seq to define 5-UTR regions and connect these molecular features to differential TE of major functional categories.

MAJOR COMMENTS:

It was shown before that differences in translational efficiency between transcripts appear to play a prominent role in stress responses or during environmental alterations. Most of the initial studies focused on determination of transcript abundances through RNA-seq and comparing it to the protein abundances through quantitative proteomics to determine TE. Recently, development of ribosomal profiling through sequencing (Ribo-Seq) allowed to determine actively translated transcripts and compare it to RNA-seq based transcriptomic measurements. For example, a recent study in *Streptomyces coelicolor* A3(2) determined the transcriptional and

translational landscape, and TSSs for transcribed sequences and associated this information with primary and secondary metabolism (Jeong et al., 2016, Nature Communications).

In this manuscript, Al-Bassam et al., take the similar approach in *C. ljungdahlii* to measure translational efficiency under different growth conditions and find that high-TE is associated with carbon and energy metabolism related genes. These claims are well-supported by the comprehensive sets of data. This is an important step forward to understand the biology of this model acetogen and provided datasets would be a great reference for future studies. However, novelty of the findings and the uniqueness of the omics approach may not be of general interest to Nature Communications audience and may benefit from publication in a more specialized journal.

We agree with the reviewer that our work has technical overlap with the Jeong et al., 2016 paper, yet it has limited analytical and conceptual overlap. We bring new insight into translational regulation by showing that translational control is widespread in metabolic pathways that are essential for growth under heterotrophic or autotrophic conditions in the industrially important bacterium, *C. ljungdahlii*. We use functional enrichment analysis to determine how the translational and transcriptional resources are being assigned to pathways/clusters in each growth condition. We demonstrate that genes in these pathways have more optimized molecular features that allow relatively higher translational efficiency. We further demonstrate that these optimized features carried by the mRNA molecule can promote higher TE when the mRNA is expressed at a higher level. We argue that rational energy allocation through mRNA optimization and translation is essential to help *C. ljungdahlii* grow on very low energy resources exemplified by autotrophic growth. These findings could be exploited to engineer/optimize transcription and translation related features in *C. ljungdahlii* pathways which could improve the yield of environmental friendly biofuels.

SPECIFIC COMMENTS:

1. There are some critical mistakes in the manuscript that makes it difficult to understand the results and manuscript requires a major proofreading. Some of these issues are listed below.

We apologize for the mistakes and we believe we have made necessary corrections in the revised version.

2. Discussion is merely a repetition of the results and does not include the overall impact of the current work for the field, connections to other studies and scientific significance. I would strongly recommend expanding on the interpretation of the results to highlight the significance and implications of the research and what needs to be done next. Also, it can benefit from the discussion of the potential technical and data analysis issues and challenges with Ribosomal profiling, TSS-seq and combining multi-omics data sets.

We have extended the discussion as recommended by reviewer #2 and we believe that we have now covered the topics that the reviewer has recommended in our discussion.

3. Reference numbers for single digit references are not formatted correctly. e.g 123 instead of 1,2,3.

We apologize for the mistakes and have made necessary corrections.

4. Figure numbers are missing in the figures.

We have made necessary corrections.

5. Page 1, L23, 29: H₂ or CO₂, “2” must be subscripted

We have made necessary corrections.

6. Page 1, L25: *Clostridium ljungdahlii* must be italic

We have made necessary corrections.

7. Page 2, L56: “..codon adaptation control TE and are crucial factors..” “and” should be removed.

This has now been removed and the paragraph was rephrased.

8. Page 2, L65: RPKM and FPKM normalization is referred to Supplementary Figure S1 but there is no information in that figure about these. Also, it is not clear how percent efficiency was calculated in Supplementary Figure S1.

We have now included the definition within the main text. In addition, Figure S1 summarizes the sequencing depth and sequencing details and how efficiency was calculated

9. Page 3, L 75-77: “We calculated the TE of each gene by dividing the translational level by the transcriptional level and noticed significant discrepancy in TE among different genes (Fig. 1b).”

Figure 1b only shows a selected region on the genome. It would be nice to have a high-level statistic on how common is this trend and also labeling for what genes are specifically highlighted in this figure would be nice.

We thank the reviewer for their suggestion. We have edited the figure to include gene names. Statistical analyses related to TE are now discussed in the last two sections.

10. In Figure 2, it is not clear how many genes are in a given subsystem.

We have now included the gene number per each subsystem in Figure 2.

Is it possible that enrichment can be biased by few genes in a given subsystem with very high RPKM or FPKM. They already see an example of this effect with “sporulation cluster (H4)”.

In this section of the result we discuss the amount of resources assigned to specific pathways, which is exactly what the figure shows. The amount of resources will be similar whether it is assigned to one gene or a group of genes in a given pathway. We agree that it is important to find out which genes are biased, therefore we now added a detailed discussion about this point in the next section and we show the details for major carbon and energy pathways in Figure 3. We also made the full dataset available in a user-friendly format (Supplementary table 2) such that researchers can rapidly find details about genes of interest.

11. Page 7, L206: “..we compared RNA-seq and Ribo-seq data of genes with low-TE (<20th percentile) or low-TE (>80th percentile).”

Do you mean or “high-TE (>80th percentile)” instead of or “low-TE (>80th percentile)”?

Yes, it is “high-TE”. We apologize for the typo and we corrected the mistake in the text.

12. Page 7, L241-243: “The difference was highly significant (Supplementary Table 2) between the two groups under all three conditions, suggesting that RBS affinity towards the initiating ribosomes is a key determinant for TE.”

Should “Supplementary Table 2” be “Supplementary Table 3”?

Yes, it was Table 3 in the first submitted manuscript, but now it is Supplementary table 5, because we had to add other necessary tables.

13. Page 9, L274-275: “Features that promote low-TE are enriched in differentially transcribed genes involved in condition-specific carbon and energy metabolic pathways”.

Do you mean “Features that promote high-TE are enriched in differentially transcribed genes involved in condition-specific carbon and energy metabolic pathways”

We have re-written this section as it wasn't clear and reanalyzed the data according to reviewer 2 and reviewer 3 recommendations.

14. Page 9, L285-287: “...carbon and energy metabolism enriched in genes that are differentially transcribed and show a low-TE, but not enriched in differentially transcribed and low-TE genes.”

Do you mean “...carbon and energy metabolism enriched in genes that are differentially transcribed and show a high-TE, but not enriched in differentially transcribed and low-TE genes.”

We have re-written this section as it wasn't clear and reanalyzed the data according to reviewer 2 and reviewer 3 recommendations.

15. Page 9, L290-292: “Group 1 represents our “test” group and consist of low-TE genes (>80th percentile) for both autotrophic growth (blue and green dots in Fig. 6a, b)”

Do you mean “Group 1 represents our “test” group and consist of high-TE genes (>80th percentile) for both autotrophic growth (blue and green dots in Fig. 6a, b)”

We have re-written this section as it wasn't clear and reanalyzed the data according to reviewer 2 and reviewer 3 recommendations.

16. On Page 9, authors calculate RAST-enrichment as the ratio of the genes in group 1 or 2 to total number of genes in a given subsystem and rank the ratios. It is not totally clear to me why they would calculate enrichment this way? I think the more appropriate way would be to calculate hypergeometric p-values, after correction for each subsystems and report significant enrichments.

Thank you for the suggestion, we have now included significant RAST categories based on Fisher exact test in Figure 6b. The significantly enriched categories are highlighted in the figure, but actual *P* values for the categories are now provided in supplementary table 6.

17. Page 10, L315-234: This paragraph and Table 1 mainly report results related to UTR features and their impact on TE. It would be more appropriate to move this paragraph to the end of the results section where authors discuss the impact of these features (Pages 6-8).

We have re-written the whole section.

18. Methods section, Data Processing: “RPKM, FPKM and correlation values of biological replicates were calculated using statistical software R (version 3.3.1).”

This is a very generic description and calculation of RPKM and FPKM are central to this paper. Therefore, authors should describe whether calculations are performed by using a specific R package or custom scripts in which case they should provide these custom scripts.

We have now included the details requested in the methods section.

19. Figure Legends, Figure 1: “RNA-seq and Ribo-seq profiles were normalized in RPM.”

What is RPM? Reads per million?

Yes it is Read per million. We have included the explicit term in the figure legend.

20. Figure Legends, Figure 2: “Differential translation and differential TE of subsystems in fructose”

Do you mean “Differential transcription and differential TE of subsystems in fructose”

It is actually differential translation. We use statistics (DESeq2 in this case) to determine significance of differential translation at the subsystem level. The transcription (middle panel with three columns) is there to allow direct comparison with the translation data and it is shown without significance calculation, as our intention is to focus on translation and TE.

Figure 5a, condition names are missing. What are the units?

We have now added the condition names and the units, which are FPKM for RNA-seq and RPKM for Ribo-seq.

Reviewer #3 (Remarks to the Author):

Article Summary:

Ribosome profiling has in recent years been used to show that there is prevalent translational regulation of gene expression under various conditions. The submitted study compares transcriptional and translational regulation of gene expression in *C. ljungdahlii*, a model acetogen organism, grown under different growth media conditions. To do this they used a combination of RNA-seq and Ribo-seq and TSS-seq (to define transcriptional start sites genome-wide). The most interesting results of the study, although perhaps not surprising given the studies on other organisms, performed in the last couple of years, is that translational control is most prevalent in those metabolic pathways that are the most useful for growth under given conditions – heterotrophic vs autotrophic growth. The authors try to explain the molecular mechanisms of translational control, which is where, in my opinion the study lacks clarity and explanatory power. All in all, this is an interesting study not only for the acetogen research community, but also for the wider community, interested in the interplay between transcription and translation. If the interpretational part of the study is improved, I would recommend it for publication.

Major issues:

1) How novel is the study? While this is, to my knowledge, the first time that translational regulation has been studied genome-wide in an acetogen, there have been other studies which

did the same for other prokaryotes and eukaryotes. These studies have also found major translational regulation. I would like the authors to do a more thorough comparison with previous studies and evaluate whether they actually found something new or are they presenting an old message on one more organism?

We think that the question about the novelty is partly due to the confusion caused by lack of clarity. We have now amended our paper and made it clearer as guided by the reviewers' comments. Apart from that, our study:

1. Provides new insights not only into the translational control of gene expression, but also sheds light on the selectivity of translational control. We showed that the translational control is highly prevalent in pathways that are most important for growth in autotrophic or heterotrophic conditions.
2. Our results show that translational efficiency is not a static constant as it varies between conditions.
3. Optimized features carried by the mRNA molecule can promote higher TE when the mRNA is expressed at a higher level. We approximate this phenomenon to enzyme kinetics.
4. It is the first paper for an anaerobic bacterium where translational control is studied comprehensively.
5. *C. ljungdahlii* is considered as a promising chassis for biofuel production, therefore it is crucial to understand how translation is specifically controlled in order to allow efficient pathway engineering. Ribosome profiling, unlike proteomics profiling, is comprehensive, relatively unbiased and specifically determines translational control. It also provides a more comprehensive coverage (same as RNA-seq) compared to proteomics.

There are many studies that have carried out shotgun proteomics and/or transcriptomics in acetogens. Transcriptomics data by itself is a poor proxy for the actual protein abundance (see also response to Reviewer #1). Proteomics data are indeed very important and are a more realistic representation of gene expression, which is protein abundance. However, proteomics has two major shortcomings. First, proteomics data (shotgun) often have poor coverage for membrane proteins^{27,28} and poor coverage for proteins with small or low abundance²⁹. Membrane proteins in particular are important for understanding energy conservation in acetogens because the Rnf complex and the ATPase (two essential energy centers) both consist of several proteins that are membrane-associated. Second, protein abundance data (from proteomics profiling) cannot resolve the extent of regulation that takes place at the translational or posttranslational level.

We have now extended the discussion such that it now addresses the recommendations from reviewer #2 and reviewer #3.

2) I find the last two sections of the results very difficult to read and understand, partly because of several typos (see below and under minor issues). In my opinion, in these sections the

authors confuse two similar, but different questions. One is, what are the features of mRNA molecules that determine whether they will be translated efficiently or not. The authors test the CAI, AU-richness of 5'UTR, strength of the RBS motif and find these all influence translational efficiency. However, these features are permanent, therefore they cannot explain why translational efficiency changes when bacteria are exposed to different growth conditions.

In this section of the results (“**TE is governed by a combination of features linked to the 5' untranslated region as well as the coding region**”), we are discussing how the 5'UTR features affect the translational efficiency in general. We do not compare between the three conditions as reviewer #3 stated. We think the reviewer might have misunderstood the point due to the many typos that we made and we apologize for this.

Basically, we show that there are many features in the mRNA molecule that can significantly affect TE. In this section we simply answer the reviewer's first question “what are the features of mRNA molecules that determine whether they will be translated efficiently or not”. The explanation of what could cause differences in TE is highlighted in the last part of the result section.

We have now re-written the last section as well. We have re-analyzed the data in order to address why some pathways have differential TE. The robustness of our conclusions is supported by high statistical significance. We now have included statistical test and have updated Fig. 6 accordingly.

Since different genes are translated efficiently in heterotrophic and autotrophic growth, one should not be looking at similarities of all high-TE genes, but at the differences between these groups – but the authors have not done so. Instead they put forward a hypothesis that (lines 282-287) “if features in the UTRs and in the coding regions have significant influence on TE in a mRNA-dependent matter, we expect to find subsystems related to carbon and energy metabolism enriched in genes that are differentially transcribed and show a low-TE, but not enriched in differentially transcribed and low-TE genes. “ This is an extremely confusing sentence. Firstly, why would there be transcriptional regulation of translation efficiency? In Figure 1, Ribo-seq vs RNA-seq correlation is shown – if there was transcriptional dependent TE changes, would not an even better correlation would have to be there for TE vs RNA-seq?

We thank the reviewer for bring up this point. We believe that the reviewer's constructive criticism helped to improve this part significantly. We are not quite sure what the reviewer meant by “but at the differences between these groups”, but this is a description of how we reanalyzed the data to address the reviewer's questions and clarify the text:

We have used the maximum TE (TE_{max} in the text) calculated for each gene from the three growth conditions. We used maximum TE as a measure of the potential affinity of mRNA towards ribosomes. We demonstrate, as the reviewer requested, that TE does increase in proportion to mRNA levels for optimized mRNA (optimized 5'UTR/coding region). Whereas for mRNA with low maximum TE, we demonstrate that the TE drops when mRNA increases and we further demonstrate that this mRNA has sub-optimized features. Furthermore, we now show how mRNA is distributed across cell functions according to its affinity towards ribosomes (maximum TE). Please see Figure 6 and Supplementary Figure 9.

Secondly, in both groups low-TE genes are used (see also lines 291 vs 293), I guess one of these should be changed to high-TE?

Yes we have corrected the text accordingly.

In the end they find an enrichment of carbon metabolism genes in the differentially expressed groups with high TE – but this should be expected already from Figure 2, where the almost same enrichment is clearly there.

We have updated this section as mentioned above (reviewer #2). We now use statistical significance (Fisher exact test) to show the enrichment.

It also does not show that TE is in any way transcriptionally dependent, just that an increase in TE correlates with differential transcription in carbon metabolism. I just don't see how any of these supports the claim that translation of these genes is affected by fluctuations in transcription (lines 312-313).

Translational buffering has been explained before. Please see Jeong *et al.*, 2016 reference 18 in the main text. We now show the dependence of TE on mRNA level and mRNA optimization of the 5'UTR and coding regions.

3) The only time the authors at least implicitly recognize that there needs to be a molecular mechanism behind the differential translation under different conditions, is in lines 251-260, where they mention the RPS1 (30 ribosome protein S1) gene.

The section where we discussed RPS1 was concerned with understanding how features in the 5'UTR and coding region of transcribed mRNA can affect TE. One of the features was the AU content of the URR. In that context, we presented RPS1 as a factor, which could influence TE via the AU content at the URR. We also cited several references that support this hypothesis. We have now extended this section and included discussion about the global RNA-binding regulator CsrA, RNaseE and IF1, IF2 and IF3. We further discuss the differential abundance of initiation factors and elongation factors and how that ratio could be important to selectively prioritize translation of optimized mRNA when expressed at higher levels. Additionally, we discussed differential translation of small ribosomal protein and other factors that lead to decreased TE.

If this gene is responsible for differential translation, then it would be interesting to find out whether it was differentially expressed in the different conditions, actually it would be interesting to find out whether any genes known to be involved in translation initiation are differentially expressed. It is possible that the molecular mechanism could be found among these genes, as was the case in other studies – see e.g. Hsieh AC, Nature 485; Thoreen CC, Nature 485 for mTOR regulated translation in mammals, which found 4E-BP, eIF4G1 and eIF4E to be the mediators of differential translation.

We thank the reviewer for this comment. There is a similar RNA-binding global regulator in *E. coli* called CsrA (carbon starvation regulator, good background information is found in uniprot: <https://www.uniprot.org/uniprot/P69913>). Its mode of action is reminiscent to that of mTOR, in

that it also regulates the translation of sizeable targets (720 transcripts). It activates the translation of the glycolysis genes in *E. coli* and represses biofilm formation, but it has no effect on the pentose phosphate pathway genes³⁰. CsrA is widely conserved in bacteria and a transcriptomics study on the *csrA* mutant has been reported in *Clostridium acytoputylicum*³¹, which demonstrated that the transcription of some central carbon metabolism genes were affected, but since CsrA is a translational regulator, we believe it is difficult to deduce solid conclusions from that paper without differential transition. A CsrA orthologue also exists in the *C. ljungdahlii* genome (*Clju_c09540*). Two small RNA molecules, CsrB and CsrC, regulate the binding activity of CsrA in *E. coli*, however no corresponding homologues are detectable in the *C. ljungdahlii* genome. The *csrA* gene is differentially up-regulated at the transcriptional level under autotrophic growth, but not at the translational level (see Supplementary table 2 for gene expression details). Since CsrA is regulated by both CsrB and CsrC, it would be difficult to determine whether CsrA regulates the translation of the glycolysis genes without the *csrA* deletion mutant and without identifying CsrB and CsrC, but it would be interesting to decipher its function in future studies, since it could a potential target to enhance biofuel production.

We have now included a discussion about differential expression of translation initiation genes and CsrA in the discussion section.

Minor issues:

4) So, one of the more interesting results of the manuscript is the selected increase in translational efficiency for metabolic genes connected to growth under heterotrophic or phototrophic conditions. Since, a metabolic reconstruction is available for *C. ljungdahlii* (ref 2), it would be interesting to see how the obtained high-TE genes match the amplitude of metabolic fluxed under heterotrophic and autotrophic growth obtained from flux balance analysis.

We thank the reviewer for this insightful suggestion. We have now compared predicted fluxes from autotrophic (H₂:CO₂) and heterotrophic (fructose) using a previously published genome-scale model²¹ by our group with experimental data. We saw 0.73 and 0.7 Pearson correlations with RNA-seq and Ribo-seq respectively when comparing at the RAST subsystem level (Supplementary Fig. 12a,b). We excluded 12 outlier subsystems (~12% of data) when we calculated the Pearson's correlation, which are highlighted in Supplementary Fig. 12.

TE and Flux had very low correlation (Pearson's R = 0.15) at the subsystem level (Supplementary Fig. 12c). Comparisons of TE and flux at the gene level were not informative and did not show any meaningful correlation (supplementary Fig. 12d). We included the details of the comparison in the discussion and the flux balance predictions in Supplementary table 7.

5) The authors use RPKM and FPKM metrics for gene expression and also used them to compare between different samples (lines 158-160). It has been shown that the TPM metric is better in this case.

We thank the reviewer for the suggestion. We followed the widely used RPKM and FPKM calculations accordingly, which we think are equally reliable. We will surely consider using the transcript per million (TPM) normalization in future research.

6) Figure 2, Supplementary Figure 3. Using the sizes and colors of circles is an interesting way of presenting a lot of information in a single figure, but this means that it needs to be explained perfectly. My interpretation is that one should look at color in columns within an experimental condition (dark red would mean that there are more transcripts, blue less), while sizes in rows and compare between conditions. This should be made even clearer in the figure legends. Also, not sure whether it's necessary to change the color map for the TE.

We edited the figure legends to make it clearer. The Ribo-seq and RNA-seq data are presented as percent values (i.e log₂ from 0-100), accordingly we chose a rainbow color map to show values as a continuum. For TE, the color map was chosen such that it is centered at zero because it is the log₂ of the ratio of Ribo-seq/RNA-seq with red color being above zero, blue color being below zero and white is zero. We therefore believe the rainbow color map is not suitable for TE.

Finally, in caption of Figure 2, the term translationally induced is used, while in Methods (Lines 470-479), significantly differentially efficient (SDE) subsystems are defined. I assume SDE and translationally induced is the same, but it would be helpful if the same terms were used throughout the manuscript.

We have removed the abbreviation.

7) Referencing within the text does not have any commas, which in some cases makes it difficult to read which references are supposed to be referenced. Example on line 72 9810.

We apologize for the typo and we corrected the mistake in the text.

8) Line 47: „we carried out cognitive analysis“. I haven't come across this term yet, what exactly is cognitive analysis?

We rephrased the sentence.

9) Lines: 116-117: „A2 and A4 had the most diff TE subsystems“. Maybe better - > „were“ the most diff TE subsystems?

Thank you for the suggestion. We changed “had” to “were” in the main text.

10) Lines 234-235: „ we compared the composition of the -10 and -35 regions of the 5' UTR by analyzing 40nt upstream of the TSS“. Should read 40 nt upstream of the start codon, see also Figure 5b.

We apologize for the mistake. We corrected the text accordingly.

11) Lines 248-249: „whereas groups with low-TE exhibit an optimal RBS motif“. Should read group with high-TE.

We have now corrected the text.

12) Line 282: missing and

We added “are” as follows: This is beneficial as these systems **are** translationally buffered.

13) Lines 457, 458: „iii) high-differentially translated genes – 90th percentile, vi) low-differentially translated genes – 10th percentile“. In the text in lines 206-207 low-TE and high-TE are described as (><20, >80%) Why are different numbers used for the same thing? . Also, vi) should be iv).

Initially, we used 10th and 90th percentiles as cutoffs to define high and low TE, but we decided to use 20th and 80th percentile, as it is more statistically robust. We forgot to update the methods part with the new cutoff values. We apologize for the mistake and the confusion. We have corrected the values and changed “vi” to “iv”.

14) While the whole section „features that promote low-TE „ is confusing, I find that especially the term low-TE is inappropriately used throughout the section (should be replaced by high-TE in several places).

We apologize for the inappropriate use of the two terms. We made changes throughout the text.

References:

1. Ingolia, N. T., Ghaemmaghami, S., Newman, J. R. S. & Weissman, J. S. Genome-wide analysis in vivo of translation with nucleotide resolution using ribosome profiling. *Science* **324**, 218–223 (2009).
2. Jeong, Y. *et al.* The dynamic transcriptional and translational landscape of the model antibiotic producer *Streptomyces coelicolor* A3(2). *Nat. Commun.* **7**, 11605 (2016).
3. Albert, F. W., Muzzey, D., Weissman, J. S. & Kruglyak, L. Genetic Influences on Translation in Yeast. *PLoS Genet.* **10**, (2014).
4. Juntawong, P., Girke, T., Bazin, J. & Bailey-Serres, J. Translational dynamics revealed by genome-wide profiling of ribosome footprints in Arabidopsis. *Proc. Natl. Acad. Sci.* **111**, E203–E212 (2014).
5. Gonzalez, C. *et al.* Ribosome Profiling Reveals a Cell-Type-Specific Translational Landscape in Brain Tumors. *J. Neurosci.* **34**, 10924–10936 (2014).
6. Lei, L. *et al.* Ribosome profiling reveals dynamic translational landscape in maize seedlings under drought stress. *Plant J.* **84**, 1206–1208 (2015).
7. Dunn, J. G., Foo, C. K., Belletier, N. G., Gavis, E. R. & Weissman, J. S. Ribosome profiling reveals pervasive and regulated stop codon readthrough in *Drosophila melanogaster*. *Elife* **2013**, (2013).
8. Liu, T. Y. *et al.* Time-Resolved Proteomics Extends Ribosome Profiling-Based Measurements of Protein Synthesis Dynamics. *Cell Syst.* **4**, 636–644.e9 (2017).
9. Belle, A., Tanay, A., Bitincka, L., Shamir, R. & O’Shea, E. K. Quantification of protein half-lives in the budding yeast proteome. *Proc. Natl. Acad. Sci.* **103**, 13004–13009 (2006).
10. Li, G. W., Burkhardt, D., Gross, C. & Weissman, J. S. Quantifying absolute protein synthesis rates reveals principles underlying allocation of cellular resources. *Cell* **157**, 624–635 (2014).
11. Dekel, E. & Alon, U. Optimality and evolutionary tuning of the expression level of a protein. *Nature* **436**, 588–592 (2005).
12. Tremblay, P., Zhang, T., Dar, S. A., Leang, C. & Lovley, D. R. The Rnf Complex of *Clostridium ljungdahlii* Is a Proton-Translocating Ferredoxin : NAD \square Oxidoreductase Essential for Autotrophic Growth. **4**, 1–8 (2013).
13. Humphreys, C. M. *et al.* Whole genome sequence and manual annotation of *Clostridium autoethanogenum*, an industrially relevant bacterium. *BMC Genomics* **16**, (2015).
14. Shin, J., Song, Y., Jeong, Y. & Cho, B. K. Analysis of the core genome and pan-genome

- of autotrophic acetogenic bacteria. *Frontiers in Microbiology* **7**, (2016).
15. Marcellin, E. *et al.* Low carbon fuels and commodity chemicals from waste gases – Systematic approach to understand energy metabolism in a model acetogen. *Green Chem.* 3020–3028 (2016). doi:10.1039/C5GC02708J
 16. Jencks, D. A. & Mathews, R. G. Allosteric inhibition of methylenetetrahydrofolate reductase by adenosylmethionine. Effects of adenosylmethionine and NADPH on the equilibrium between active and inactive forms of the enzyme and on the kinetics of approach to equilibrium. *J. Biol. Chem.* **262**, 2485–2493 (1987).
 17. Schomburg, D., Salzmann, M. & Stephan, D. in (eds. Schomburg, D., Salzmann, M. & Stephan, D.) 95–99 (Springer Berlin Heidelberg, 1994). doi:10.1007/978-3-642-78521-4_16
 18. Lau, E. *et al.* A large dataset of protein dynamics in the mammalian heart proteome. *Sci. Data* **3**, (2016).
 19. Lau, E. *et al.* Integrated omics dissection of proteome dynamics during cardiac remodeling. *Nat. Commun.* **9**, (2018).
 20. Boisvert, F.-M. *et al.* A Quantitative Spatial Proteomics Analysis of Proteome Turnover in Human Cells. *Mol. Cell. Proteomics* **11**, M111.011429 (2012).
 21. Nagarajan, H. *et al.* Characterizing acetogenic metabolism using a genome-scale metabolic reconstruction of *Clostridium ljungdahlii*. *Microb. Cell Fact.* **12**, 118 (2013).
 22. Fu, X. *et al.* Identification of Osr2 Transcriptional Target Genes in Palate Development. *J. Dent. Res.* **96**, 1451–1458 (2017).
 23. Lee, A.-R., Kim, J.-H., Cho, E., Kim, M. & Park, M. Dorsal and Ventral Hippocampus Differentiate in Functional Pathways and Differentially Associate with Neurological Disease-Related Genes during Postnatal Development. *Front. Mol. Neurosci.* **10**, (2017).
 24. Wang, L. *et al.* GCN5 Regulates FGF Signaling and Activates Selective MYC Target Genes during Early Embryoid Body Differentiation. *Stem Cell Reports* (2017). doi:10.1016/j.stemcr.2017.11.009
 25. Maroni, G. *et al.* Prep1 prevents premature adipogenesis of mesenchymal progenitors. *Sci. Rep.* **7**, (2017).
 26. Aklujkar, M., Leang, C., Shrestha, P. M., Shrestha, M. & Lovley, D. R. Transcriptomic profiles of *Clostridium ljungdahlii* during lithotrophic growth with syngas or H₂ and CO₂ compared to organotrophic growth with fructose. *Sci. Rep.* **7**, (2017).
 27. Chandramouli, K. & Qian, P.-Y. Proteomics: Challenges, Techniques and Possibilities to Overcome Biological Sample Complexity. *Hum. Genomics Proteomics* **2009**, 1–22 (2009).
 28. Beck, M., Claassen, M. & Aebersold, R. Comprehensive proteomics. *Current Opinion in Biotechnology* **22**, 3–8 (2011).
 29. Choong, W. K. *et al.* Informatics View on the Challenges of Identifying Missing Proteins from Shotgun Proteomics. *J. Proteome Res.* **14**, 5396–5407 (2015).
 30. Sabnis, N. A., Yang, H. & Romeo, T. Pleiotropic regulation of central carbohydrate metabolism in *Escherichia coli* via the gene *csrA*. *J. Biol. Chem.* **270**, 29096–29104 (1995).
 31. Tan, Y., Liu, Z.-Y., Liu, Z., Zheng, H.-J. & Li, F.-L. Comparative transcriptome analysis between *csrA*-disruption *Clostridium acetobutylicum* and its parent strain. *Mol. Biosyst.* **11**, 1434–1442 (2015).

Reviewers' comments:

Reviewer #2 (Remarks to the Author):

In response to reviewer critics, Al-Bassam et al., significantly improved the manuscript titled "Optimization of Carbon and Energy Utilization Through Differential Translational Efficiency". They have performed additional experiments such as shotgun proteomics to determine protein levels and additional data analysis to support main conclusions of the manuscript. Supporting tables and figures are also appended into the revised submission. Authors have addressed most of the questions and criticism through their response document, and edited the manuscript to reflect necessary changes. Most of my criticism of the study is addressed yet I still would like to point to few of my previous issues that might further improve the manuscript.

1. Authors responded to the criticism of reviewers # 2 and #3 about the novelty of the study with a list of points that highlight the merit of the study. I agree that all these points are exciting and important. However, they do not provide an evaluation of the findings in comparison to existing studies to satisfactorily address if these messages have been reported before. A thorough comparison of the main findings with literature and discussion of its novelty would clarify the significance of this study.

2. In response to improving the Discussion, Al-Bassam et al., made major revisions and this section has been significantly improved. In parallel to the above point, including a comparative assessment of the novelty will further improve the Discussion.

3. Authors addressed the potential bias for genes with a very high RPKM or FPKM for the functional enrichment by saying "The amount of resources will be similar whether it is assigned to one gene or a group of genes in a given pathway." While this statement is true, I still think it will matter if you are calculating the enrichment score for a given pathway if that pathway has only few genes assigned with very high RPKM or FPKM. A further explanation might clarify this point.

Minor issues:

1. Supplementary Figure 1: "dot" (.) is used in the table as a 1000 separator instead of comma.

Reviewer #3 (Remarks to the Author):

Since the original publications, the authors have added a considerable amount of work to improve the analysis of their data, as well as improved their interpretation of the results. There are still a few minor issues that need to be addressed, otherwise I'm very happy with the second iteration of the manuscript.

Minor comments:

The authors have added a proteomics part and have performed FBA analysis according to recommendations, which is commendable. However, the methods used are not presented in the main part of the paper, but only in a supplementary file. This is in principle OK, but then at least the supplementary methods should be referenced somewhere (and I could not find them). Also, the FBA methods, especially the reduction of the model could be described in more detail (only reference is given).

there are some issues with supplementary table numbering and referencing, eg I could not find the referenced Table S4, but there is an extra Table S7, while there is not Table S5?

line 417: delete the "the"

Reviewer #4 (Remarks to the Author):

Al-Bassam et al have addressed the regulation of key enzymes involved in acetogenesis by *Clostridium ljungdahlii* by a transcriptome and Ribo-seq approach. This reviewer has not seen the previous version. The paper is rather lengthy and not pleasant to read. It culminates in two major points:

1. A multi-omics approach to study regulation of certain enzymes.

This approach is not new, it confirms previous studies. Furthermore, the approach is not without problems since a real proteomics is not done. The authors have addressed this point in their rebuttal and argue that they don't have the methodology established and presented a shotgun proteomics. However, this is not convincing. Anyway, the approach used is only confirmatory.

2. The authors make a long story about enzymes of the heterotrophic and autotrophic life style that they have identified to be regulated. I am afraid to say that a simple analysis of the literature would have shown that this question has been satisfactorily addressed in certain acetogens, including *C. ljungdahlii*. It is well known, that enzymes of the WLP are upregulated somewhat during autotrophic growth. This was established by transcriptomics, proteomics, and, most important, by measuring enzyme activities. The latter is of course the major read out for regulation. From this part of the study, nothing new can be learned.

In sum, the study is only confirmatory in methodology and conclusions.

Reviewer #5 (Remarks to the Author):

The authors addressed most of the issues raised by the Reviewer 1. Regarding Question 4, however, the authors' response is not enough to clear the original Question.

For RNA-seq, correlations between two replicates (Pearson's correlations: 0.995 (CO), 0.991 (Fru), and 0.989 (H₂/CO₂)) are acceptable for using further analysis, as like the previously reported works. For Ribo-seq, however, correlations between two replicates (Pearson's correlations: 0.995, 0.952, and 0.930 (0.926 was presented in S_Fig.2; which one is correct?)) are not reliable. Especially, Ribo-seq (RPKM) could hinder the accurate calculation of TE (very important parameter in this study), because of the low-quality of Fru (0.952) and H₂/CO₂ (0.926) samples. To clearly support the authors' conclusion, additional replicates for Ribo-seq should be included.

Optimization of Carbon and Energy Utilization Through Differential Translational Efficiency

Second Response to reviewers' comments

We would like to thank all reviewers for their time and their valuable comments, which we genuinely believe has improved the manuscript significantly.

Reviewer #2

In response to reviewer critics, Al-Bassam et al., significantly improved the manuscript titled "Optimization of Carbon and Energy Utilization Through Differential Translational Efficiency". They have performed additional experiments such as shotgun proteomics to determine protein levels and additional data analysis to support main conclusions of the manuscript. Supporting tables and figures are also appended into the revised submission. Authors have addressed most of the questions and criticism through their response document, and edited the manuscript to reflect necessary changes. Most of my criticism of the study is addressed yet I still would like to point to few of my previous issues that might further improve the manuscript.

We are grateful to the reviewer for the suggestions and we are glad that the reviewer found the second iteration has improved significantly.

1. Authors responded to the criticism of reviewers # 2 and #3 about the novelty of the study with a list of points that highlight the merit of the study. I agree that all these points are exciting and important. **However, they do not provide an evaluation of the findings in comparison to existing studies to satisfactorily address if these messages have been reported before. A through comparison of the main findings with literature and discussion of its novelty would clarify the significance of this study.**

2. In response to improving the Discussion, Al-Bassam et al., made major revisions and this section has been significantly improved. In parallel to the above point, including a comparative assessment of the novelty will further improve the Discussion.

As the reviewer knows, the literature in this field is quite vast and because of limitations set by the journal, we can only reference what is most relevant to the work presented in our manuscript. We referenced work that directly relates to 5'UTR/coding region effect on TE (REFs 27, 37, 38, 43, 44 and 45), anti-Shine Dalgarno (REFs 40, 46), and AU rich region upstream of RBS (REFs, 31, 32 and 33) throughout the text. Further, we guide the reader towards review articles (REF 37, 38 and 39) available on this topic. Furthermore, we have now modified our discussion to specifically highlight how our study compared to existing work and what new findings and insights into translational control is gained by our work. The revised text now reads:

“We took a systems-level approach to understand the overall TE and how it is linked to transcription and *cis*-acting regulatory sequences at the 5'UTR. The rate by which TE increases in response to increased mRNA levels depends mainly on the 5'UTR region. Highly optimized 5'UTRs are significantly enriched in growth condition-specific carbon and energy metabolic pathways, whereas suboptimal 5'UTRs are enriched in housekeeping genes. This prudent assignment of optimized 5'UTRs to carbon and energy pathways ensures faster translational response of urgently required pathways, which is vital when scarce resources are transiently available. By the same token, the assignment of suboptimal 5'UTRs to housekeeping genes (lower TE) ensures stable translation as well as the use of minimal resources to sustain maximal growth. Differential TE of metabolic pathways and genes has been reported previously for both eukaryotes¹ and prokaryotes². The effect of the 5'UTR length and secondary structure on TE and their role in regulation of secondary metabolite translation has been reported in *Streptomyces coelicolor*³. However, the effect of 5'UTR and RNA expression levels on TE and how this strategy is used to allocate resources efficiently was previously unknown. Here, we illustrate how the interplay between defined features in the 5'UTR and RNA transcription levels determines condition-specific TE of metabolic pathways. Moreover, we show how *C. ljungdahlii* modulates the TE levels for metabolic pathways in a growth condition-dependent manner. Our work unravels how acetogens utilize the differential TE mechanism to use carbon and energy resources optimally to thrive at the thermodynamic energy limit of life.”

Furthermore, our quantitative analysis pipeline can be readily applied in the future to analyze new and existing Ribo-seq and RNA-seq datasets to unravel how energy resource allocation is managed in different bacteria.

3. Authors addressed the potential bias for genes with a very high RPKM or FPKM for the functional enrichment by saying “The amount of resources will be similar whether it is assigned to one gene or a group of genes in a given pathway.” While this statement is true, I still think it will matter if you are calculating the enrichment score for a given pathway if that pathway has only few genes assigned with very high RPKM or FPKM. A further explanation might clarify this point.

We believe that our calculation is clear, intuitive, and based on strong statistical grounds. We have added a paragraph in the methods section to explain how we measured the enrichment in order to clarify this point:

“We calculated the total number of sequencing reads that were aligned to each RAST subsystem, regardless of whether genes in a given subsystem are differentially regulated or not. We then calculated the RPKM/FPKM for each subsystem and reported the values of the top 20 subsystems that were significantly differentially transcribed/translated according to DESeq2. We assumed that the amount of resources required for transcription/translation will be similar whether it is assigned to one gene or a group of genes in a given RAST subsystem.”

Minor issues:

1. Supplementary Figure 1: “dot” (.) is used in the table as a 1000 separator instead of comma.

Thank you. We have now corrected the “dot” to a “comma”.

Reviewer #3

Since the original publications, the authors have added a considerable amount of work to improve the analysis of their data, as well as improved their interpretation of the results. There are still a few minor issues that need to be addressed, otherwise I'm very happy with the second iteration of the manuscript.

We are grateful to the reviewer for the suggestions and we are glad that the reviewer found the second iteration has improved significantly.

Minor comments:

The authors have added a proteomics part and have performed FBA analysis according to recommendations, which is commendable. However, the methods used are not presented in the main part of the paper, but only in a supplementary file. This is in principle OK, but then at least the supplementary methods should be referenced somewhere (and I could not find them). Also, the FBA methods, especially the reduction of the model could be described in more detail (only reference is given).

Thank you for the comment. We had reached the maximum citations (70) in the main text and since the methods needed additional citations for the sake of clarity, we were forced to include them (together with proteomics methods) into a supplementary methods file. We are sorry that we missed to refer to the methods.

We have now included the necessary references to the supplementary methods and the reduction of the model is now described in detail as:

“The amplitude of metabolic fluxes was determined using FVA⁴. Reactions carrying an absolute flux lower than 10 were set to zero. All reactions that could not carry flux under autotrophic and heterotrophic conditions were identified and removed from the models⁵. To uniformly sample the solution space of iNH637, optGpSampler for MATLAB with Gurobi Optimizer Version 6.5.0 was used⁶. The reduced models were sampled with 50,000 sample points.”

there are some issues with supplementary table numbering and referencing, eg I could not find the referenced Table S4, but there is an extra Table S7, while there is not Table S5?

Thank you for pointing this out. Supplementary Table 4 is actually referred to in lines 208 and 209. However, we renamed it now “Supplementary Table 3” as it occurs in the main text before Supplementary Table 3 which is now renamed to “Supplementary Table 4”. All Supplementary Tables are now correctly accounted for. We apologize for the confusion. We highlighted the changes in blue font in the supplementary methods document.

line 417: delete the "the"

Thank you, we deleted “the”.

Reviewer #5 (Remarks to the Author):

The authors addressed most of the issues raised by the Reviewer 1. Regarding Question 4, however, the authors’ response is not enough to clear the original Question.

For RNA-seq, correlations between two replicates (Pearson’s correlations: 0.995 (CO), 0.991 (Fru), and 0.989 (H2/CO2)) are acceptable for using further analysis, as like the previously reported works. For Ribo-seq, however, correlations between two replicates (Pearson’s correlations: 0.995, 0.952, and 0.930 (0.926 was presented in S_Fig.2; which one is correct?) are not reliable. Especially, Ribo-seq (RPKM) could hinder the accurate calculation of TE (very important parameter in this study), because of the low-quality of Fru (0.952) and H2/CO2 (0.926) samples. To clearly support the authors’ conclusion, additional replicates for Rio-seq should be included.

We understand the concern raised by Reviewer #5 regarding the fructose and the H2/CO2 Ribo-seq experiments. However, we are certain that the data are of “high-quality” and that two replicates suffice for the following reasons:

1. Ribosome profiling can answer many questions regarding translation. Some of which require high-resolution and extremely high reproducibility. For example, analyzing the exact positions of the ribosome (the P and A sites) and the codon reading frames inside each coding region will introduce a significant amount of noise associate with this type of analysis⁷. For such high-resolution analyses, more biological replicates and extremely high correlations would be beneficial. However, since our manuscript is mostly a systems-level (global level) analysis of translational efficiency under three different conditions, the correlation between duplicates is actually very high for all conditions (Pearson 0.97, 0.97 and 1.00, **Fig. A**). When Spearman correlations are implemented, which is the accepted standard when reporting reproducibility⁷, the correlations are almost perfect (Spearman 0.99, 0.99, 1.00, **Fig. A**). We also want to highlight that we did not remove any outliers in **Fig. A**, which is also true for the gene-level Ribo-seq correlations shown in the manuscript (**Supplementary Fig. 2a**). The slight drop in correlation at the gene level does not affect the validity of our analysis and conclusions.

2. There are several manuscripts published in *Nature Communications* or other higher impact journals^{3,8,9,10}, which have similar or even lower correlation than the ones reported here. All of these studies use biological duplicates. Nevertheless, we performed additional analyses to assess the quality of our Ribo-seq replicates and the reliability of the data. We used the DESeq2 algorithm to determine the standard errors and estimate the dispersion between conditions for Ribo-seq data. **Fig. B** shows the distribution of standard error (SE) between two conditions (fructose vs H₂:CO₂ and fructose vs CO) for the Ribo-seq data. In brief, the SE indicates how far the sample mean deviates from the population mean. Genes relevant to our study (depicted in **Fig. 3**) are highlighted in blue in **Fig. B**.

As the standard error is a valid measure of variability regardless of the distribution, we used the 95th percentile as a threshold to select genes with high SE. We only found three genes (*Clju_RS12830*, *Clju_RS12835*, *Clju_RS14145*, encoding for oxidoreductase, PTS fructose transporter subunit IIB and, Ni/Fe hydrogenase, respectively) that are potentially involved in pathways presented in **Fig. 3** (**these genes are not shown in Fig. 3 because they are not differentially translated**), which showed SE above the threshold. However, none of these genes are relevant to our study, since *Clju_RS12830*, *Clju_RS12835* are hardly expressed and *Clju_RS14145* is only expressed at a very low level (maximum Ribo-seq RPKM=0.4, and maximum RNA-seq FPKM=3.1, see **Supplementary Table 2**). The high SE shown for these three genes can be justified by the low number of read counts presented in all conditions. Genes with low read counts can present strong variance of logarithmic fold change (common metrics for relative expression or ratio). It is a direct consequence of count data (sequencing data), in which ratios are strongly biased when counts are low.

DESeq2 is an algorithm that can robustly handle as few as two replicates per condition¹¹. DESeq2 assumes that genes with similar expression levels have similar dispersion and the dispersion for each gene is estimated using maximum likelihood estimation (black dots - **Fig. C**). A linear model is fitted to capture the overall trend of dispersion-mean dependence (blue line - **Fig. C**). It is expected that different genes will have different scales of biological variability, but overall, the estimated dispersion will follow a reasonable distribution. Genes that do not fit to the model are detected as dispersion outliers (points circled in orange - **Fig. C**). None of the genes discussed in our manuscript and highlighted in **Fig. 3** were detected as dispersion outliers considering all the six Ribo-seq experiments (three conditions).

We have now added **Fig A** as **Supplementary Fig. 2b** in the revised version of the manuscript and referred to it in the main text. Given this analysis, the very high quality of our data, the fact that Ribo-seq data are standardly reported as duplicates, and that the data analysis for genes and subsystems highlighted in our manuscript will not change by adding an additional replicate, we are certain that the data quantity of our study suffice the conclusions.

Figure A. Pearson and Spearman correlations between Ribo-seq replicates at the RAST subsystem level in the three growth conditions used. The minimum Pearson correlation is 0.97 and the minimum Spearman correlation is 0.99.

Figure B. Standard error distributions between two experimental conditions. Orange bars indicate the standard error of all *C. ljungdahlii* genes. Blue dashes indicate the genes discussed in the manuscript. Dashed red line indicates the 95th percentile threshold.

Figure C. Estimation of dispersion. Gene-wise maximum likelihood estimate (MLEs) were obtained using the gene's data (black dots). A curve (blue) is fit to the MLEs to capture the overall trend of dispersion-mean dependence. The black points circled in orange are detected as dispersion outliers.

1. Dieudonné, F. X. *et al.* The effect of heterogeneous Transcription Start Sites (TSS) on the translome: Implications for the mammalian cellular phenotype. *BMC Genomics* (2015). doi:10.1186/s12864-015-2179-8
2. Taylor, R. C. *et al.* Changes in translational efficiency is a dominant regulatory mechanism in the environmental response of bacteria. *Integr. Biol. (United Kingdom)* (2013). doi:10.1039/c3ib40120k
3. Jeong, Y. *et al.* The dynamic transcriptional and translational landscape of the model antibiotic producer *Streptomyces coelicolor* A3(2). *Nat. Commun.* **7**, 11605 (2016).
4. Mahadevan, R. & Schilling, C. H. The effects of alternate optimal solutions in constraint-based genome-scale metabolic models. *Metab. Eng.* **5**, 264–276 (2003).
5. Zuniga, C. *et al.* Predicting dynamic metabolic demands in the photosynthetic eukaryote *Chlorella vulgaris*. *Plant Physiol.* pp.00605.2017 (2017). doi:10.1104/pp.17.00605
6. Megchelenbrink, W., Huynen, M. & Marchiori, E. optGpSampler: An improved tool for uniformly sampling the solution-space of genome-scale metabolic networks. *PLoS One* **9**, (2014).
7. Diamant, A. & Tuller, T. Estimation of ribosome profiling performance and reproducibility at various levels of resolution. *Biol. Direct* (2016). doi:10.1186/s13062-016-0127-4
8. Rooijers, K., Loayza-Puch, F., Nijtmans, L. G. & Agami, R. Ribosome profiling reveals features of normal and disease-associated mitochondrial translation. *Nat. Commun.* (2013). doi:10.1038/ncomms3886
9. Rutkowski, A. J. *et al.* Widespread disruption of host transcription termination in HSV-1 infection. *Nat. Commun.* (2015). doi:10.1038/ncomms8126
10. Bazzini, A. A., Lee, M. T. & Giraldez, A. J. Ribosome Profiling Shows That miR-430 Reduces Translation Before Causing mRNA Decay in Zebrafish. *Science* **336**, 233–237 (2012).
11. Love, M. I., Anders, S. & Huber, W. *Differential analysis of count data - the DESeq2 package.* *Genome Biology* **15**, (2014).

REVIEWERS' COMMENTS:

Reviewer #2 (Remarks to the Author):

After reviewing their responses to this round of critics I believe manuscript updates and responses from Al-Bassam et al., satisfied my additional concerns. They have updated manuscript to reflect the uniqueness of their study and clarified the concerns for functional enrichment. In addition, they addressed minor issues raised by Reviewer#3.

Reviewer#5 requested additional replicates in response to criticism for correlation between Ribo-seq experiments. While I understand the concerns of the reviewer and believe additional replicates will strengthen the conclusions, I think conclusions in the manuscript are justified given the additional amount of data, new set of analysis in the response document and use of similar number of replicates in previous studies cited. However, I would recommend to revise the language on lines 94 and 95 in the manuscript "The high reproducibility suggests that the addition of more replicates will not increase statistical robustness significantly" and provide the potential caveats Reviewer raised for correlation of Ribo-seq results with two replicates.

Reviewer #3 (Remarks to the Author):

I am satisfied with the improvement to the manuscript and the responses of the authors. I recommend that the article be published.

Reviewer #5 (Remarks to the Author):

The authors cleared my question by adding SI Fig. 2a, newly, which could support their conclusion more concretely.

Optimization of Carbon and Energy Utilization Through Differential Translational Efficiency

Third Response to Reviewers' comments

We would like to thank all reviewers and the editorial committee for their efforts and their constructive comments and criticisms. We are delighted that the reviewers are satisfied with our revised version.

Reviewer #2 (Remarks to the Author):

After reviewing their responses to this round of critics I believe manuscript updates and responses from Al-Bassam et al., satisfied my additional concerns. They have updated manuscript to reflect the uniqueness of their study and clarified the concerns for functional enrichment. In addition, they addressed minor issues raised by Reviewer#3.

Reviewer#5 requested additional replicates in response to criticism for correlation between Ribo-seq experiments. While I understand the concerns of the reviewer and believe additional replicates will strengthen the conclusions, I think conclusions in the manuscript are justified given the additional amount of data, new set of analysis in the response document and use of similar number of replicates in previous studies cited. However, I would recommend to revise the language on lines 94 and 95 in the manuscript "The high reproducibility suggests that the addition of more replicates will not increase statistical robustness significantly" and provide the potential caveats Reviewer raised for correlation of Ribo-seq results with two replicates.

Thank you for the suggestion, we have changed the text in order to avoid overstating and it now reads:

"The high correlations between biological replicates reflects the high reproducibility of our data."

Reviewer #3 (Remarks to the Author):

I am satisfied with the improvement to the manuscript and the responses of the authors. I recommend that the article be published.

Thank you.

Reviewer #5 (Remarks to the Author):

The authors cleared my question by adding SI Fig. 2a, newly, which could support their conclusion more concretely.

Thank you.